# Alpha- and Beta-Coronaviruses in Humans and Animals: Taxonomy, Reservoirs, Hosts, and Interspecies Transmission

**DOI:** 10.3390/microorganisms14010043

**Published:** 2025-12-24

**Authors:** Bekbolat Usserbayev, Kuandyk Zhugunissov, Izat Smekenov, Nurlan Akmyrzayev, Akbope Abdykalyk, Khayrulla Abeuov, Balnur Zhumadil, Aibarys Melisbek, Meirzhan Shirinbekov, Samat Zhaksylyk, Zhanerke Nagymzhanova, Ainur Seidakhmetova, Chiara Beltramo, Simone Peletto, Aslan Kerimbaev, Sergazy Nurabaev, Olga Chervyakova, Nurlan Kozhabergenov

**Affiliations:** 1Research Institute for Biological Safety Problems, National Holding QazBioPharm, LLP, Gvardeisky 080409, Kazakhstan; k.zhugunisov@biosafety.kz (K.Z.); n.akmyrzayev@biosafety.kz (N.A.); a.abdykalyk@biosafety.kz (A.A.); kh.abeuov@biosafety.kz (K.A.); balnur.zhumadil@bk.ru (B.Z.); a.melisbek@biosafety.kz (A.M.); m.shirinbekov@biosafety.kz (M.S.); s.zhaksylsyk@biosafety.kz (S.Z.); zh.nagymzhanova@biosafety.kz (Z.N.); aynur.1990gvardeisk@mail.ru (A.S.); a.kerimbayev@biosafety.kz (A.K.); s.nurabayev@biosafety.kz (S.N.); o.chervyakova@biosafety.kz (O.C.); n.kozhabergenov@biosafety.kz (N.K.); 2Scientific Research Institute of Biology and Biotechnology Problems, al-Farabi Kazakh National University, Almaty 050040, Kazakhstan; smekenovizat@gmail.com; 3Istituto Zooprofilattico Sperimentale del Piemonte, Liguria e Valle d’Aosta, Via Bologna 148, 10154 Torino, Italy; chiara.beltramo@izsplv.it

**Keywords:** coronavirus, respiratory infection, natural reservoir, intermediate host, receptor, One Health, interspecies transmission

## Abstract

The *Coronaviridae* family represents a broad group of RNA-containing viruses that infect humans and animals. This family belongs to the order *Nidovirales* and is divided into four main genera: α-CoV, β-CoV, γ-CoV and δ-CoV. It is particularly noteworthy that representatives of β-CoV have caused serious epidemics in humans, such as the outbreaks of SARS-CoV, MERS-CoV, and COVID-19 caused by SARS-CoV-2. Although the clinical manifestations of CoVs can range from mild cold-like symptoms to severe respiratory diseases, they share common features in their structure, modes of transmission, and natural reservoirs. Identifying natural reservoirs, as well as establishing intermediate hosts, is crucial for understanding the mechanisms of interspecies transmission of CoVs. These processes are often mediated by molecular interactions between viral spike (S) proteins and cellular receptors of different species, which contribute to zoonotic outbreaks. Thus, the interaction of various species and the study of these processes of viral spread, cross-species transmission, and pathogen evolution play a key role in ensuring global biological safety. Therefore, we conducted this review to summarize the data from existing studies focused on the taxonomy of CoVs, their main types, natural reservoirs, intermediate hosts, pathways of interspecies transmission, and the significance of the One Health concept as an interdisciplinary approach to monitoring, prevention and control of CoV infections at the intersection of human, animal, and environmental health. We examined databases such as PubMed, Science Direct, Web of Science, and Google Scholar to identify relevant scientific articles in English available for such a review. The aim of this work is to study the taxonomy and classification of coronaviruses, as well as to identify their natural reservoirs, intermediate hosts, and applicable control measures. A review of human and animal coronaviruses has revealed their evolutionary diversity, their main natural reservoirs, their intermediate hosts, and their interactions with cellular receptors. This information allows for a better understanding of the mechanisms by which the viruses are transmitted from animals to humans. The concept of One Health demonstrated the interconnections between human, animal and environmental factors.

## 1. Introduction

While the 20th century played a pivotal role in shaping strategies for the control acute respiratory diseases, particularly those caused by influenza viruses, by the second decade of the twenty-first century it had become clear that the emergence and rapid spread of novel coronavirus infections presented a major global health challenge [1,2].

Coronaviruses (CoVs) are enveloped, positive-sense RNA viruses distinguished by prominent club-shaped surface spikes. They possess some of the largest genomes among RNA viruses and employ a unique mechanism of replication [3]. CoVs are widespread pathogens capable of infecting a broad range of animal species, including both birds and mammals. They can cause severe and often fatal diseases affecting the respiratory, gastrointestinal, cardiovascular, and nervous systems. Their broad host range makes them particularly concerning [4].

Consequently, the health risks associated with CoVs remain both persistent and significant. Investigating their emergence and origin, as well as identifying primary and intermediate hosts, is a critical scientific priority. A comprehensive understanding of these factors will support effective control and prevention strategies, ultimately promoting economic stability and strengthening global public health systems. Unlike previous reviews, which have mainly focused on SARS-CoV-2, in this review we combine human and veterinary perspectives, detailing natural reservoirs and intermediate hosts, and discussing «One Health» issues.

The aim of this study is to examine the taxonomy and classification of CoVs, and to identify their natural reservoirs, intermediate hosts, and applicable control measures.

## 2. Methods

The search across all databases (PubMed, Science Direct, Web of Science, Google Scholar, etc.) was conducted using the following keywords and terms: «coronavirus», «alpha-coronavirus», «beta-coronavirus», «SARS-CoV-2», «MERS-CoV», «bats», «One Health», etc. The selection of terms was carried out according to thematic sections. Articles published between YEAR and YEAR in English were considered. Priority was given to peer-reviewed original studies and reviews, especially those addressing taxonomy, host range, receptor interactions, and zoonotic transmission. As a result, both original and review studies were selected. Particular attention was paid to publications in English whose titles, abstracts and key topics met the selection criteria. The selected materials were analyzed in terms of their relevance to the objectives of this review.

## 3. Taxonomy and Classification of Coronaviridae

The scientific classification of CoVs began in the late 1960s. Viruses such as infectious bronchitis virus (IBV), mouse hepatitis virus (MHV), B814, Human coronavirus 229E (HCoV-229E), and Human coronavirus OC43 (HCoV-OC43) were grouped together and designated as CoVs based on their electron microscopic and morphological features. This led to the development of a simplified taxonomic system. Initially, these viruses were assigned to the *Myxovirus* group. The classification proposal was submitted by A.P. Waterson, who at the time led the Myxovirus Study Group within the International Committee on Taxonomy of Viruses (ICTV) [3]. A brief timeline of the evolution of CoV classification is shown in Figure 1.

In August 2018, the taxonomic classification of the *Coronaviridae* family was revised [3]. The hierarchical structure at the genus level was updated, and a new subgenus rank was introduced [7]. Subgenera are defined by threshold values of patristic distances on a phylogenetic tree, constructed using the maximum likelihood method based on multiple alignment of the complete genomes of all known members of the family [8]. The threshold values were established as 0.186 for the subgenus, 0.789 for the genus, and 1.583 for the subfamily level [3].

The family *Coronaviridae* is currently classified into three subfamilies based on genetic, serological, and phylogenetic criteria: *Letovirinae*, *Pitovirinae*, and *Orthocoronavirinae.* These subfamilies belong to the suborder *Cornidovirineae* within the order *Nidovirales.* The third subfamily, *Orthocoronavirinae*, includes four genera *Alphacoronavirus* (α-CoV), *Betacoronavirus* (β-CoV), *Deltacoronavirus* (δ-CoV), and *Gammacoronavirus* (γ-CoV), 25 subgenera, and 52 recognized species [9] (Figure 2).

As shown in Figure 2, CoV species are widely distributed pathogens that infect a variety of animal species, including birds and mammals [3]. Among them, α-CoVs and β-CoVs primarily infect mammals, while δ-CoVs and γ-CoVs are capable of infecting both mammals and a broad range of avian species.

## 4. Genomic Organization and Structural Proteins

Electron microscopic studies indicate that CoV virions exhibit a spherical morphology, with sizes ranging from 80 to 220 nm [10]. The CoV virion is a complex structure that is integral to the pathogenesis of diseases associated with these viruses. The primary structural proteins constituting the virion are glycoprotein (S), envelope proteins (E), membrane proteins (M), and nucleocapsid proteins (N) (Figure 3) [1,10]. The primary structural protein of the virus, glycoprotein S, is essential for binding to host cell receptors—a crucial step in the infection process [11]. The M protein is important for virus assembly and for maintaining its stable shape, thus ensuring the structural integrity of the viral envelope [12]. The phosphorylated nucleocapsid protein N safeguards viral RNA and facilitates its replication within host cells [13]. The E protein plays essential roles in viral assembly and in the pathogenesis of infection [14]. Certain CoVs also contain an additional envelope protein exhibiting both haemagglutination and esterase (HE) activities [5,10].

The CoV genome is a linear RNA of positive polarity, ranging in size from 27,000 to 32,000 nucleotides and encoding approximately 22 to 29 proteins [20]. Its organization includes the following elements: a 5′ leader, untranslated region (UTR), replicase, the structural proteins S, E, M, and N, a 3′ UTR, a poly(A) tail, and a set of accessory genes integrated within the structural genes near the 3′ terminus [10]. The 5′ cap and 3′ poly(A) tail structures enhance mRNA translation efficiency, which is essential for the synthesis of replicase polyproteins. The replicase gene encodes non-structural proteins (NSPs) that participate in genome replication and modification of the host cellular environment, thereby supporting viral survival and propagation [21,22]. This gene contains two large overlapping open reading frames (ORF1a and ORF1b), which are critical for viral replication [1]. In addition to these, the genome includes the genes for structural proteins S, E, M, and N, interspersed with a variable number of additional ORFs, highlighting the complexity and variability of CoV infections [1,10]. Figure 4 illustrates this organization of the genome and highlights the importance of all its components.

## 5. Alpha- and Beta-Coronaviruses of Animals

### 5.1. Porcine Coronaviruses

Pigs serve as significant natural reservoirs for various CoVs. They cause considerable economic harm to agriculture and possess epizootological and potential zoonotic significance. Currently, six distinct CoVs are recognized as infecting pigs (Table 1). Four of these viruses are classified within the genus α-CoV: transmissible gastroenteritis coronavirus (TGEV), porcine respiratory coronavirus (PRCV), porcine epidemic diarrhea virus (PEDV), and porcine acute diarrhea syndrome coronavirus (SADS-CoV). The remaining porcine coronaviruses are classified into distinct genera: one is part of the β-CoV genus (porcine haemagglutination encephalomyelitis virus- PHEV), while another is categorized under the δ-CoV genus (porcine *deltacoronavirus*—PDCoV). TGEV, PRCV, and PHEV have been established in porcine populations for decades, while PEDV, PDCoV, and SADS-CoV are regarded as emerging CoVs [24].

An analysis of current literature indicates that porcine CoVs primarily impact the digestive tract and lead to acute epizootics in young pigs. Some CoVs, including PRCV and PHEV, impact both the respiratory and central nervous systems, illustrating the variability in tropism and pathogenesis among different genera of CoVs within the same animal species.

### 5.2. Feline Coronaviruses

Domestic cats (*Felis catus*) serve as natural hosts for certain CoVs. Some strains lead to benign intestinal infections, while others result in severe systemic diseases, including feline infectious peritonitis (FIP) [34,35]. Cats may also be vulnerable to zoonotic coronaviruses, such as Severe Acute Respiratory Syndrome 2 (SARS-CoV-2) [36].

Table 2 illustrates that FCoVs exhibit a spectrum of pathogenicity, from mild intestinal ailments to lethal systemic illnesses. Significant focus has been directed towards FIP, a mutational variant of FECV that induces severe immunopathology [34].

### 5.3. Canine Coronaviruses (CCoVs)

Canine coronavirus (CCoV) was initially identified in Germany in 1971. This virus, impacting canines, disseminated swiftly across the globe, encompassing areas such as Asia, Europe, South America, and North America [38]. A.W. McClurkin et al. demonstrated that members of the Canidae family are susceptible to infection by the TGEV coronavirus. CCoV induces respiratory and intestinal diseases in canines, rendering it a significant focus of investigation and surveillance within veterinary medicine [39]. This also presents a significant health risk to both wild and domestic animals.

Canine respiratory coronavirus (CRCoV) was identified in the United Kingdom in 2003, representing a subset of CoVs that impact dogs [40]. CRCoV exhibits a broad distribution across North America, Japan, and various European nations. This virus induces respiratory infections in canines, particularly within kennel populations [41]. Molecular genetic studies indicate that CRCoV exhibits low genetic similarity to CCoV in terms of protein S and RdRp, while demonstrating a high degree of similarity to Bovine Coronavirus (BCoV) [40].

### 5.4. Bovine Coronaviruses (BCoVs)

In 1972, bovine coronavirus (BCoV) was recognized as a causative agent of respiratory and intestinal infections in cattle and wild ruminants [3,42,43]. Respiratory BCoV is linked to mild respiratory illnesses, including cough and rhinitis, as well as more severe conditions such as pneumonia in calves aged 2 to 6 months. BCoV is commonly identified in nasal secretions, pulmonary tissues, and the gastrointestinal tract, including fecal matter. This highlights its capacity to induce both respiratory and intestinal symptoms, thereby posing a considerable health risk to young cattle. BCoV is widely present in global cattle populations, as indicated by the seroprevalence of antibodies to this virus [42].

BCoVs, akin to CoV, have been identified in various ruminants, including multiple species of deer (*Cervidae*), water goats (*Kobus*), giraffes (*Giraffa*), alpacas (*Lama pacos),* black antelopes (*Hippotragus niger),* and buffalo (*Syncerus caffer*) [44].

### 5.5. Mink Coronavirus (MCoV)

Epizootic catarrhal gastroenteritis (*ECG—Epizootic catarrhal gastroenteritis*) in mink was initially documented in 1975. This disease was documented in the USA, Canada, Scandinavia, the People’s Republic of China, and the former USSR, resulting in considerable harm to the mink population [45]. The disease exhibits a seasonal pattern, primarily affecting minks aged 4 months and older, with an approximate morbidity rate of 100% and a mortality rate of less than 5% [3,40].

In the early 1980s, the infectious characteristics of ECG were documented. Following the identification of viral particles in the feces of American minks during an ECG epizootic in Denmark in 1985, the designation ECGV (ECG virus) was introduced [3]. Infected minks exhibit anorexia syndrome, leading to deterioration in condition and pelts, ultimately resulting in economic losses for mink producers. Vlasova A. N. et al. noted that, to date, CoV detected in ECG has not been isolated or sequenced [45].

### 5.6. Ferret Coronaviruses (FRCoVs)

In 1993, a novel intestinal disease affecting domestic ferrets (*Mustela putorius furo*) was recognized in the USA, characterized by clinical symptoms including infectious diarrhea [46]. Summary of CoV infections in ferrets is presented below (Table 3).

Table 3 illustrates that ferret coronaviruses present a diverse array of clinical manifestations, ranging from localized intestinal infections to systemic lesions. The investigation of CoVs in ferrets is significant for both veterinary medicine and the comprehension of interspecies transmission and viral adaptation mechanisms, including those related to SARS-CoV-2. Ferrets serve as sensitive models and natural hosts for certain viruses, rendering them important subjects in the fields of virology and infectious disease research [49]. Laboratory studies have highlighted the susceptibility of ferrets to SARS-CoV-2, raising concerns due to their confirmed vulnerability to human coronaviruses [50]. Furthermore, ferrets were utilized in laboratory settings as models for pathogenesis studies and vaccine development related to COVID-19 during the pandemic [49].

### 5.7. Equine Coronavirus (ECoV)

Equine coronavirus (ECoV), which shares antigenic similarities with BCoV, was first identified in 1999 through serological analysis of feces and intestinal tissues from a foal exhibiting diarrhea; however, efforts to isolate the virus were unsuccessful. This study, along with previous research, has identified CoVs or virus-like particles in foals and adult horses suffering from intestinal disease. However, the pathogenicity of these viruses and their causal relationship with the onset of intestinal disease is not well elucidated. Additional research is required to ascertain the prevalence of ECoV infection and evaluate its role as a contributor to intestinal disease in horses [51].

### 5.8. Beluga Whale Coronavirus SW1

In 2008, a 13-year-old male beluga whale succumbed to a disease marked by generalized lung damage and acute liver failure at a zoo in San Diego, CA, USA [3]. Histological, electron microscopic, and molecular genetic analyses were conducted to identify the etiological agent of the disease. Histological analysis demonstrated severe, multifocal, and confluent centrilobular-massive acute liver necrosis. Electron microscopic analysis identified numerous spherical viral particles measuring approximately 60–80 nm, accompanied by nuclei of about 45–50 nm within the cell cytoplasm; however, this observation alone was inadequate for a comprehensive characterization of the virus. Molecular genetic studies identified the etiological agent of this disease as a novel coronavirus, designated SW1 (San Diego whale 1) [3,52].

### 5.9. Coronaviruses in Hedgehogs (Erinaceus spp.)

In 2014, a novel species of CoV, designated HdCoV-1 (Hedgehog coronavirus 1), was identified through a molecular genetic analysis of biological samples collected from a hedgehog population in northern Germany, conducted by the University of Bonn [53]. The virus infects the intestinal mucosa and, upon excretion with feces, exhibits a yellow or green coloration without presenting additional clinical symptoms [3]. Corman V.M. et al. conducted a detailed analysis of the hedgehog gut, revealing that the virus exhibited the highest concentration in samples from the lower gastrointestinal tract. This aligns with viral replication in the lower intestine and fecal-oral transmission [53].

## 6. Human Alpha- and Beta-Coronaviruses

The documented history of human coronaviruses (HCoV) begins in the mid-1960s of the twentieth century. Tyrrell D.A. and Bynoe M.L. were the first to describe HCoVs, which were isolated from patients suffering from acute respiratory diseases (ARD) [3,8,54].

### 6.1. Human Coronavirus-229E (HCoV-229E)

HCoV-229E, a coronavirus, was initially identified in humans in 1966 [55]. It is classified under the genus α-CoV and the subgenus *Duvinacovirus* [9]. Liu D.X. et al. indicated that HCoV-229E correlated with cold symptoms in healthy adults; however, they observed that children and the elderly exhibited greater susceptibility to lower respiratory tract infections [55]. The incubation period for the HCoV-229E virus ranges from 2 to 5 days, while the duration of the illness may extend from 2 to 18 days. The primary symptoms consist of headache, rhinorrhea, sneezing, pharyngitis, and overall malaise [54].

### 6.2. Human Coronavirus—OC43 (HCoV-OC43)

The HCoV-OC43 coronavirus was first isolated in 1967 in Salisbury, England, from patients exhibiting cold symptoms. For virological studies, it was initially cultured in human mesenteric epithelium, embryonic trachea, and nasal tissue cultures [56]. HCoV-OC43 is typically linked to mild upper respiratory tract infections; however, research indicates its potential neuroinvasive properties. Moreover, HCoV-OC43 is capable of inducing persistent infections within human neural cell structures [55]. The incubation period is between 2 and 4 days. HCoV-OC43 can transmit in public settings via coughing and sneezing, leading to mild upper respiratory tract infections in adults [56].

The human coronaviruses HCoV-229E and HCoV-OC43, recognized during that period, were deemed sufficiently safe for administration to volunteers without restrictions. Regrettably, aside from strains OC43 and 229E, several of the initially identified human coronaviruses have not persisted to the present day [3].

### 6.3. Severe Acute Respiratory Syndrome Coronavirus (SARS-CoV)

SARS-CoV is the causative agent of an epidemic first identified in November 2002 in Guangdong Province, China, fundamentally altering the perception of human coronaviruses, which had previously been regarded as relatively benign [57,58,59]. The SARS outbreak of 2002–2003 affected over 29 countries globally, leading to about 8000 cases and 774 fatalities [60]. The virus is transmitted through airborne droplets at close proximity [61], and in certain instances, direct and indirect contact with respiratory secretions, feces, or animal vectors may facilitate transmission [62,63].

The first symptoms of the disease include fever (>38 °C), typically accompanied by alternating chills and trembling, along with additional manifestations such as headache, fatigue, and myalgia. The incubation period of SARS typically ranges from 2 to 7 days [64], occasionally extending up to 10 days [63]. Initially, the sickness manifests as minor respiratory symptoms in certain cases, while some patients exhibit signs of diarrhea during the early stages of fever. Within 3 to 7 days, the illness presents in the lower respiratory phase, characterized by the onset of cough and dyspnoea, which may subsequently lead to or exacerbate hypoxaemia [64]. Disease severity and mortality fluctuate with age, with the highest mortality rate (over 50%) in those over 65 years and the lowest in those aged 0 to 24 years [63].

### 6.4. Human Coronavirus-NL63 (HCoV-NL63)

In 2004, a novel coronavirus species, HCoV-NL63, was identified from clinical specimens of a 7-month-old child [65,66,67] and adults suffering from acute respiratory infections in the Netherlands [55,68,69]. The principal indicators of sickness are characterized by symptoms including fever, cough, and rhinorrhea, followed by the onset of pneumonia symptoms (mild to moderate respiratory illness) [67]. The identification of HCoV-NL63 in multiple regions globally, including Canada, Australia, Belgium, China, and Japan, demonstrates its global dissemination [69,70,71,72,73,74,75,76,77,78].

### 6.5. Human Coronavirus-HKU1 (HCoV-HKU1)

In 2005, a novel strain of HCoV (HCoV-HKU1) was discovered from an infected patient at Hong Kong University [3]. The symptoms of HCoV-HKU1 infection include rhinorrhea, cough, nasal congestion, fever, sputum production, pharyngodynia, chills, nasal discharge, and tonsillar hypertrophy. Liu D.X. et al. report that around fifty percent of patients infected with HCoV-HKU1 have febrile convulsions, circulating concurrently with respiratory syncytial virus, with epidemics typically preceding the influenza season. HCoV-HKU1 is globally ubiquitous, with higher infection rates observed in adults compared to other age groups [55].

### 6.6. Middle East Respiratory Syndrome (MERS-CoV)

The first case of MERS was documented in Jeddah, Saudi Arabia, in 2012 [70,71]. The clinical manifestations of MERS-CoV infection vary from asymptomatic or mild respiratory symptoms to severe acute respiratory disease, which may be deadly [72,73,74,75,76,77,78,79,80,81]. The typical incubation time of MERS-CoV ranges from 5 to 7 days, while documented cases indicate it can extend from 2 to 14 days [73]. Since the initial report of MERS-CoV, there have been 2200 documented cases, resulting in 858 fatalities [74]. Several nations, including Algeria, Austria, Bahrain, the People’s Republic of China, Egypt, France, Germany, Greece, the Islamic Republic of Iran, Italy, Jordan, Kuwait, Lebanon, Malaysia, the Netherlands, Oman, the Philippines, Qatar, the Republic of Korea, the Kingdom of Saudi Arabia, Thailand, Tunisia, Turkey, the United Arab Emirates, the United Kingdom, the United States of America, and Yemen, have notified the World Health organization of the identification of MERS-CoV within their borders, in compliance with international health regulations [72,75].

MERS-CoV is a complex infection characterized by various transmission pathways. Prior research indicates that zoonotic transmission from camels to humans occurs via direct or indirect contact with infected one-humped camels, highlighting the necessity of animal health management and surveillance of camel populations [76,77]. Conversely, human-to-human transmission primarily transpires among close contacts and within healthcare environments [72,76,78].

### 6.7. SARS-CoV-2

In late December 2019, several healthcare institutions in Wuhan, Hubei Province, PRC, began documenting instances of pneumonia of indeterminate origin. These incidents were associated with a wholesale market that dealt in seafood and livestock. On 31 December 2019, the China Centre for Disease Control and Prevention (China CDC) deployed a task force to assist local health authorities in Hubei Province and Wuhan City in conducting an epidemiological and etiological investigation. Through the analysis of biological samples from patients utilizing molecular genetics and virology techniques, the virus was fully identified and its genome sequenced. The illness caused by this novel coronavirus was initially termed New Coronavirus Pneumonia (NCIP) [79]. On 12 January 2020, the World Health organization (WHO) officially named the disease Coronavirus Disease 2019 (COVID-19), derived from Coronavirus Disease 2019. Subsequently, the ICTV classified the pathogen as a coronavirus associated with severe acute respiratory syndrome and named it SARS-CoV-2 [80].

The virus is transmitted via the respiratory tract of an infected individual, including through coughing, sneezing, and even minute moisture droplets expelled during respiration [81]. The virus can disseminate among family members, friends, and colleagues in close proximity, necessitating particular vigilance in scenarios when personal protective equipment is absent. SARS-CoV-2 infection exhibits a spectrum of symptoms, ranging from moderate respiratory issues to severe outcomes, including pneumonia and mortality [82]. Typically, the duration from virus infection to symptom manifestation is 5–6 days, although it may extend to 14 days [81].

Since its initial identification, SARS-CoV-2 has spread globally, emerging as a significant public health crisis. As per the WHO, by the conclusion of the first quarter of 2020, the virus had impacted a minimum of 114 nations, with over 118,000 recorded cases and around 4000 fatalities. In light of this situation, the WHO designated the COVID-19 outbreak as a pandemic resulting from SARS-CoV-2 on 11 March 2020 [83]. As of 9 November 2025, the WHO reports 778,922,858 confirmed cases globally, with over 7,103,341 resulting in fatalities [84]. The initial cases of infection with the novel CoV in Kazakhstan were documented on 13 March 2020 [85], involving two individuals who had returned from Germany [86]. In compliance with constitutional regulations, quarantine measures were implemented in the country from 16 March to 15 April 2020 to safeguard the population [87].

## 7. Natural Reservoirs and Intermediate Hosts of CoV

### 7.1. The Origin of Some Animal Coronaviruses

The investigation of the origins and transmission routes of CoVs in natural ecosystems is essential for understanding zoonotic transmission mechanisms and preventing future outbreaks [88,89,90]. Animals play a central role in the spread of CoVs due to their extensive genetic diversity, ecological adaptability, and capacity to host cross-species viral transmission. They may act both as natural reservoirs and as intermediate hosts facilitating the propagation of the virus [90,91].

To support a comprehensive epidemiological assessment, a comparative table (Table 4) was compiled that summarizes key information on various CoV species, including their taxonomic classification, primary hosts, and known reservoirs. These data reflect the current scientific understanding of the role of animals in the ecology and interspecies transmission of CoVs.

As shown in Table 4, the animal CoVs presented are of considerable taxonomic diversity, spanning several genera of the family *Coronaviridae*, including α-CoV and β-CoV. CoVs are characterized by a high degree of host-specificity [93], but also show the capacity for interspecies transmission, especially in close contact between animals and humans [105]. CoVs found in pigs, raccoons and hedgehogs are of greatest concern from an epidemiological point of view. CoVs affecting pigs pose a serious economic threat to agriculture, while viruses detected in ferrets, minks [106], raccoons [101] and hedgehogs [102,103,104] require continuous epidemiological and epizootological surveillance. Continuous monitoring and research help to identify risks in advance and develop effective measures to prevent them. This, in turn, is crucial for preventing possible disease outbreaks and protecting animal and human health.

### 7.2. Origins of HCoVs

Animal-derived bacteria and viruses are the most prevalent and responsible for lethal human diseases [106]. Ye Z.W. et al. discovered that animals may serve as intermediate hosts for HCoV solely if they harbour viruses exhibiting significant genetic similarity at elevated genomic levels [89]. These primordial viruses are host-adapted and non-pathogenic, establishing the reservoir host as a stable and enduring host for HCoV. If HCoV initially infects an intermediate host immediately prior to or during human infection, it is inadequately adapted and subsequently becomes pathogenic. This intermediate host acts as a zoonotic reservoir for human infection, enabling the virus to temporarily proliferate and disseminate among people, hence augmenting the scale of human infections. If the intermediate host is unable to maintain onward transmission, HCoV may lead to a dead-end infection; however, if the host undergoes successful adaptation, it may become the principal natural reservoir of the virus [89]. A recent study indicated that all HCoVs are of zoonotic origin; specifically, SARS-CoV, MERS-CoV, HCoV-NL63, and HCoV-229E likely derive from bats, whereas HCoV-OC43 and HCoV-HKU1 possibly originate from rodents [107] (Figure 5).

Figure 5 describes the existing scientific data regarding the origin and transmission pathways of CoVs from other species to humans. Coronaviruses HCoV-229E and HCoV-NL63 are alpha-coronaviruses classified under the subgenera *Duvinacovirus* and *Setracovirus*, respectively [9]. Their genesis is proposed to be associated with bats (*Chiroptera*), specifically the species Triaenops afer and members of the genus *Hipposideros* [110,111,112]. The HCoV-229E coronavirus may have employed one-humped camels (*Camelus dromedarius*) as intermediary hosts in its transmission from bats to humans [113]. Moreover, supplementary research indicates that alpacas could function as potential intermediate hosts. This broadens the array of possible intermediate hosts within the camelid family [114]. Simultaneously, transmission of HCoV-NL63 is believed to have transpired via an unnamed intermediary host [113].

The coronaviruses HCoV-OC43 and HCoV-HKU1 are classified within the Embecovirus subgenus of the β-CoV genus [9]. Research indicates that the primary natural reservoirs of these two HCoVs are rodents [113], particularly rats (*Rattus norvegicus)* and mice (*Mus musculus*) [115,116]. This indicates that the virus may have been transmitted from rodents to humans through potential intermediary hosts. Research indicates that the HCoV-OC43 may have infiltrated the human population in the late 19th century via interspecies transmission from cattle [56]. This hypothesis is founded on genomic research of the virus, demonstrating its affinity to coronaviruses (BCoV) that impact animals [117]. Unlike most HCoVs, with the exception of HCoV-NL63, HCoV-HKU1 has yet to have a definitive intermediate host identified as of now [112]. Transmission to humans may occur by direct or indirect contact with infected rodents, as well as through a contaminated environment [118]. Fundamental evidence about the zoonotic transmission of the virus in public settings indicates that it spread several decades or centuries ago, thereafter adapting solely to the human population [112,117].

SARS-CoV, MERS-CoV, and SARS-CoV-2 are recognized as extremely dangerous CoVs responsible for global epidemics and pandemics. Research indicates that the natural reservoirs of the SARS-CoV, MERS-CoV, and SARS-CoV-2 viruses are classified as members of the genus *Rhinolophus* [119,120], family *Vespertilionidae* [73,121,122], and species within the genus *Rhinolophus* among bats, respectively [123,124,125]. The research cited has demonstrated that bats serve as natural reservoirs for highly deadly CoVs. This discovery implies that the transmission of CoVs from bats to people likely occurs indirectly via intermediary hosts. Research indicates that civets (*Paguma larvata*) [101,126], one-humped camels (*Camelus dromedarius*) [89,127,128], and pangolins (*Manis javanica*) [129,130] served as intermediary hosts in the transmission of SARS-CoV, MERS-CoV, and SARS-CoV-2 viruses to humans, respectively.

Dimonaco N.J. and colleagues observed in their work that the genetic similarity between pangolin viruses and bat CoVs is comparatively low in the context of SARS-CoV-2 [131]. This indicates that there is no compelling evidence to regard pangolins as a direct vector for viral transmission to humans. Consequently, researchers persist in examining the concept of an intermediary vector that may significantly contribute to the transmission of SARS-CoV-2. Comprehending the mechanisms of viral transmission is essential to avert such pandemics in the future. Nonetheless, the definitive intermediate host remains ambiguous [132].

SARS-CoV-2 may have originated in nature through a multistep zoonotic process. Bats serve as a crucial natural reservoir, whereas the intermediary host that enabled transmission to humans may have been an inadequately characterized mammal or other animal species. The definitive establishment of the epidemiological link necessitates further field and laboratory investigations, along with enhanced wildlife surveillance.

## 8. Receptor Usage and Molecular Determinants of Host Range

Numerous investigations indicate that the term «coronavirus» pertains to a characteristic of its protein coat, which exhibits projecting spikes. These spikes give the viral particles the shape of a crown, which is where the name comes from the Latin word corona, meaning crown [3,133]. This structure is crucial for the virus’s capacity to adhere to and infect cells. The S protein of CoVs is the most critical component in this structure, as it is essential for the virus’s entry into the cell [134]. Initially, it attaches to a receptor on the cell surface, subsequently fusing the viral and cell membranes to facilitate entry [133]. This mechanism enables the virus to infiltrate the cell and initiate its replication [1]. The initial and crucial stage in the infection of host cells is the identification of receptors by the virus [133].

The *S* protein includes an ectodomain, a transmembrane anchor, and a brief intracellular tail (Figure 6) [133,135]. The ectodomain comprises two components: the S1 subunit, which interacts with the receptor, and the S2 subunit, which is embedded in the membrane. S1 binds to the host cell receptor, while S2 facilitates the union of the viral and cellular membranes [92]. The S1 subunit comprises two domains: an N-terminal domain (S1-NTD) and a C-terminal domain (S1-CTD), also referred to as the S1 C-domain. Certain papers indicate that one or both domains of the S1 subunit may serve as a receptor-binding domain (RBD) [133,135]. The binding contacts between the receptor-binding domain (RBD) of the CoV and its receptor are critical determinants of the virus’s host species and the potential for interspecies transmission [90]. Furthermore, a component of the viral S1 protein facilitates the recognition of host receptors, encompassing protein and carbohydrate structures, which is crucial for viral entrance into cells [135].

Thakur S. and colleagues describe that any changes in the structure of the SARS-CoV-2 spike (S) protein inevitably affect its virulence and pathogenicity. Although most emerging mutations generally reduce these characteristics or prove detrimental to the virus, the S glycoprotein remains the most antigenic component, making it the centre of evolutionary transformations. It is precisely here that adaptive mutations arise, capable of increasing transmissibility, enhancing infection efficiency, and contributing to immune evasion [136]. Below (Table 5) are frequently observed mutations in the S protein across various SARS-CoV-2 variants.

Majumdar P. and colleagues note that mutations are most often observed in the S1 subunit of the spike (S) protein, and that recently almost half of the amino acid residues in the receptor-binding domain (RBD) have undergone mutations. This, in turn, creates significant challenges in the development of antiviral drugs and vaccines. Moreover, these mutations affect the stability of the spike protein, its affinity for the receptor, as well as its sensitivity to neutralizing monoclonal antibodies (mAb) and convalescent serum. Thus, studying the identified mutation patterns and their impact on viral pathogenesis is crucial for understanding the evolution of spike protein antigenicity in the context of combating the virus [145].

### 8.1. Receptor Diversity of Some Human and Animal Coronaviruses

#### 8.1.1. Aminopeptidase N (APN)

Aminopeptidase N (APN) is a transmembrane Zn^2+^ aminopeptidase and type II protein located in the gut, neurological system, dendritic cells, and monocytes of several hosts [105,146,147].

Coronaviruses that attach to APNs employ their S1 CTD domain. The interaction mechanism between RBD and APN varies across distinct coronaviruses. These variations may influence the capacity of viruses to engage with host cells and ascertain the likelihood of interspecies transmission [105,148].

#### 8.1.2. Angiotensin-Converting Enzyme 2 (ACE2)

Angiotensin-converting enzyme 2 (ACE2), discovered only in the 2000s, is widely expressed and found in various human organs and tissues, including the lungs and extrapulmonary tissues [149]. Its extensive distribution throughout several tissues significantly contributes to its interaction with viruses. This renders it a crucial receptor for the entry of viruses, particularly SARS-CoV and SARS-CoV-2, into cellular structures [105]. This receptor is a homologue of the ACE enzyme, including 805 amino acids. It features a single extracellular N-terminal domain with an active catalytic site, a C-terminal membrane anchor, and a conserved zinc-binding domain HEXXH [150].

#### 8.1.3. Sialic Acid

Sialic acid is a monosaccharide that coats the surfaces of eukaryotic cells, creating a protective barrier. It is found on glycoproteins and glycolipids, contributing to intercellular interactions [151]. Viruses actively utilize sialic acid for host cell recognition, since it functions as a crucial receptor for numerous diseases, including parainfluenza, rotaviruses, adenoviruses, polyomaviruses, CoVs, and influenza viruses [105].

#### 8.1.4. Dipeptidyl Peptidase 4 (DPP4)

Dipeptidyl peptidase 4 (DPP4), first characterized in 1966, is often referred to as CD26 T-cell activation antigen or adenosine deaminase binding protein (ADBP). DPP4 modulates various cellular processes, including adherence to the extracellular matrix, proliferation, and significantly influences T cell development and function. These characteristics render it a crucial molecule for numerous biological and immunological functions [152]. It is extensively expressed in multiple organs and tissues, including the lung, liver, intestine, immune cells, and kidney [105,153]. The following data offers an overview of the receptors utilized by several coronaviruses, encompassing both human and animal strains (Table 6).

As can be seen in Table 6, different CoVs show diversity in their choice of cellular receptors, allowing them to infect a wide range of vertebrates. The most versatile and frequently encountered receptors are aminopeptidase N (APN), ACE2, sialic acids and Dipeptidyl peptidase 4. These molecules, in turn, are key factors in the process of viral entry into cells and in determining the mechanisms of interaction between different types of coronaviruses and host cells.

The receptor landscape of CoVs illustrates their significant flexibility, facilitating interspecies transmission of the virus. The continued presence of receptors like APN, ACE2, and DPP4 across several animal species facilitates the virus’s circulation and establishes circumstances for its transmission from natural reservoirs to humans. This versatility is crucial for comprehending the virus’s evolution and the mechanisms by which it adapts to new hosts. The examination of receptor specificity is crucial for comprehending viral adaptation, evolution, and forecasting potential outbreaks.

### 8.2. Interspecies Transmission of CoV

Recent research suggests that CoVs can infect a diverse array of hosts, encompassing many animals and humans. α-CoVs and β-CoVs are primarily located in mammals, γ-CoVs in avians and certain marine animals, while δ-CoVs are most frequently observed in wild birds and swine [40]. CoVs can transcend interspecies barriers by modifying tissue tropism and adapting to various ecological niches, hence enhancing their prevalence and stability. The receptor-binding domain (RBD), situated within the S gene of the CoV genome, may identify identical receptors across many animal species. The virus’s contact with the receptor, alongside evolutionary selection, frequent homologous recombination, and point mutations, enables it to surmount interspecific barriers and swiftly adapt to a new host [154,155].

CoVs can traverse species barriers, enabling them to infect many animal species. This results from multiple factors, including a high mutation rate that facilitates viral adaptation, alterations in the *S* gene (which encodes a fusion protein) that enable the virus to engage with cells from various species, and a substantial RNA genome (~29,903 nucleotides) that heightens the probability of mutations and recombination, culminating in the emergence of novel strains. CoVs can infect their primary host as well as inadvertently infect other animal species, including dogs, cats, and pigs, leading to cross-species transmission and the emergence of novel illnesses. This indicates the existence of mutant variants adapted to various species [154].

BCoVs exemplify a phenomenon that transcends species limits. Bovine CoVs (BCoV-like) are classified among the β-CoV of the subgenus Embecovirus [9] and are significant pathogens for both domesticated and wild animals [156]. A comparative characterization of the principal representatives of BCoV-like CoVs is provided below (Table 7).

Consequently, as illustrated in Table 7, BCoV-like CoVs encompass a broad host range, including domestic cattle, goats, sheep, wild ruminants, and exotic species such as giraffes. Clinical symptoms vary from diarrhea and enteritis to respiratory illness. Molecular genetic research indicates that the genome of BCoV-like CoVs shares significant similarities with those of other animal CoVs. Investigating their epidemiology is essential to avert interspecies transmission and to formulate effective management strategies in both agricultural and natural settings. To date, there is no molecular genetic analysis of CoVs specific to caribou/deer (*Cervidae*) Sika deer and Muskox (*Ovibox moschatus*). Therefore, we did not compare the homology of the genome with other CoVs.

Amer H.M. states in his paper that there are presently six species of camelids, categorized into two tribes: Camelini (Old World camels) and Lamini (New World camels) [156]. The Old World camels comprise two species: the dromedary (*Camelus dromedarius*) and the Bactrian camel (*Camelus bactrianus*). Approximately 90 percent of all Old World camels are dromedaries. Dromedaries (one-humped camels) are natural reservoirs of the MERS-CoV and act as a conduit for transmission to humans and other domesticated animals [167]. A comparative description of the primary representatives of CoVs is provided below (Table 8).

Table 7 indicates that camelid coronaviruses have been detected in both domestic and wild animals. The primary clinical manifestation is diarrhea. Molecular genetic studies indicate that the genomes of camelid CoVs exhibit significant resemblance to the genomes of other species, including HCoVs.

Moreover, research indicates that CCoV, FCoV and porcine coronaviruses can recombine, suggesting their concurrent existence inside the same host organism. Genetic interchange between early strains of CCoV and FCoV (CCoV-I and FCoV-I) with an unidentified coronavirus produced two novel viral variants, CCoV-II and FCoV-II. Sequence analysis verifies that the TGEV virus originated via interspecies transmission of CCoV-IIfrom an infected canine (Figure 7) [170].

Parkhe P. et al. suggest that the method of virus-cell interaction also affects the formation of recombinant strains, as previously mentioned [154]. The amino acid makeup of the APN receptor in human, feline, and porcine CoVs exhibits 78% identity, signifying substantial genetic similarities among these viruses. Feline APN can act as a receptor for various CoVs, including FIPV, FECV, TGEV, CCoV, and HCoV-229E. Based on this information, felines can contract TGEV, CCoV, or HCoV-229E [154].

Consequently, CoVs can be transferred among multiple animal species, facilitating their dissemination while posing risks to their hosts.

## 9. One Health Aspects and Global Surveillance

### 9.1. Future Perspectives on Coronavirus Infections in the Context of One Health

In the last twenty years, humanity has encountered significant challenges due to coronaviruses, ranging from SARS in 2002 to the COVID-19 pandemic [171]. The COVID-19 pandemic presented an unparalleled challenge that highlighted the necessity for a comprehensive strategy in the prevention and management of zoonotic illnesses. The One Health concept is a widely recognized multidisciplinary paradigm [172]. Furthermore, Horefti E. delineates five principal domains of action within the One Health framework: food safety, human–animal communication, antimicrobial resistance of microorganisms, water contamination prevention and control, and the management of zoonoses, which are infections transmitted from animals to humans [173]. Consequently, One Health seeks to amalgamate medical, veterinary, and environmental initiatives to avert and eradicate emerging disease epidemics. The fifth and last reason is that the mentioned professionals have been charged with responding rapidly during epidemics and monitoring and mitigating public health concerns related to zoonoses [172].

Horefti E. et al. highlighted in their study that the WHO declared a global pandemic due to the appearance and dissemination of the SARS-CoV-2 virus, detailing its transmission between humans and animals during the earliest phases of virus containment [173]. The author indicated in his paper that the virus emerged during an inopportune period, specifically in winter when the influenza virus was proliferating in Wuhan. He noted that the air in the market, where various animals were sold, was excessively humid, that meat was sold alongside the animals, and that numerous individuals frequented the market to purchase these products. This implies that viruses can traverse species barriers, potentially resulting in the establishment of new disease foci.

Ghai R.R. et al. articulated that the formation of a federal interagency coordination committee for COVID-19 One Health in the United States would constitute a significant advancement in securing a unified and effective response to a pandemic [93]. This strategy enhanced interagency communication, expedited updates, and coordinated essential messages, resulting in a more efficient response.

For prevention of any disease that presents a public health risk, it is essential to operate inside the One Health framework, rather than solely relying on the pertinent authorities. Shehata A.A. et al. contend that the effective and comprehensive implementation of the One Health concept necessitates the involvement of diverse stakeholders, including social health specialists, physicians, veterinarians, epidemiologists, pharmaceutical corporations, vaccinologists, government officials, and economists [105]. They ought to collaborate on clinical case identification, laboratory diagnosis, and epidemiological surveillance of the virus, disease management, enhancement of quarantine protocols, treatment, and immunization of the populace. In this context, several publications delineate approaches designed to prevent the transmission of illnesses and safeguard public health (Figure 8) [173].

Figure 8 illustrates that the One Health idea offers multiple strategies to avert perilous infections of public significance that jeopardize human health, daily activities, and the living environment. A holistic approach is seen here: encompassing individual personal hygiene, diagnosis, treatment, and prevention of diseases, with collaborative efforts between society and the state in combating infections.

The One Health perspective is crucial and seeks to establish a comprehensive approach to global health. In the forthcoming years, this notion will be pivotal in examining the interactions among animals, humans, and the environment, mitigating new zoonotic diseases, addressing antibiotic resistance, and formulating eco-friendly solutions [172,173,176]. Moreover, coordination between international organizations and national health systems will be enhanced, emphasizing epidemic preparedness, the influence of external variables on health, and the management of global threats [177,178].

A significant area is the advancement of antiviral medications, vaccines, and diagnostic assays. This domain is crucial for the prevention and efficient management of infections. Furthermore, quick adaptability platforms and international collaboration are critical components in combating the significant pathogen. The established platforms can swiftly spread information regarding a new virus, while worldwide collaboration can facilitate the coordination of international research and clinical trials [179].

### 9.2. The Effect of Global Warming

Several studies have highlighted the role of climate change in shaping the ecology and transmission dynamics of infectious diseases. Xu et al. reported that global warming alters wildlife habitats and migration patterns, thereby modifying the distribution of viruses, bacteria, and parasites carried by migratory species [180]. Using infectious disease surveillance data from wild birds and bats—major reservoirs for numerous zoonotic pathogens—collected between 1962 and 2020, the authors showed that many viruses are expanding northward across continents, whereas bacterial spread appears less directly correlated with climate change [180].

Other authors have emphasized that greenhouse gas (GHG) emissions not only intensify climate-related hazards but may also influence the transmission of high-risk pathogens. Although the potential impact of climate change on infectious diseases is widely acknowledged, the extent of increased human vulnerability remains insufficiently characterized. Mora et al. further demonstrated that global climate change may amplify infectious disease risks worldwide by interacting with multiple climate hazards (Figure 9) [181].

A systematic assessment of more than 77,000 scientific publications identified approximately 830 instances in which climate factors affected epidemic spread. Among 375 infectious diseases evaluated over several decades, 218 (58%) were directly influenced by warming. Rising air temperatures were associated with 160 diseases, increased precipitation with 122, and flooding with 121. Transmission in many of these cases involved vectors such as insects, rodents, or bats [181].

According to the United Nations Environment Programme (UNEP), climate change also disrupts water resource dynamics. Warming contributes to permafrost degradation, increased flooding, and the expansion of wetlands, conditions that favour mosquito proliferation and the transmission of vector-borne diseases including malaria and other tropical fevers. Elevated river basin temperatures may facilitate cholera outbreaks, while warmer seasons promote tick activity and associated tick-borne diseases. Furthermore, climate-driven human migration and urban expansion may enhance the spread of infectious diseases [182]. Overall, climate change represents a multifactorial driver of disease emergence and spread, underscoring the need for coordinated international strategies and the integration of One Health approaches.

### 9.3. SARS-CoV-2 in the Context of One Health: From Potential Emergence to Genomic Monitoring

CoVs exhibit a significant ability for interspecies transmission and genetic recombination. The emergence of SARS-CoV (2002), MERS-CoV (2012), and SARS-CoV-2 (2019) substantiates the notion that these viruses, considered to originate from animals, were transferred to humans through intermediate hosts [105,107]. At the onset of the 21st century, subsequent to the detection of SARS-CoV and its epidemiological ramifications, numerous scientific organizations commenced rigorous investigations into animal CoVs to evaluate their zoonotic potential-namely, their capacity for transmission to humans. The South China Wet Seafood Market in Wuhan, Hubei Province, PRC [183], is regarded as the primary origin of SARS-CoV-2 [132]. This market features a diverse array of mammalian species that offer both seafood and live wild animals, fostering close interactions among wild and domestic animals, as well as humans [123]. Upon transmission of the virus to an intermediate host, SARS-CoV-2 could experience adaptive genetic recombination, facilitating its proliferation among the population [132]. The main host of SARS-CoV-2 is likely bats (*Rhinolophus affinis*), as the significant genetic resemblance between SARS-CoV-2 and CoVs obtained from bats indicates that this novel virus may have also arisen from bats [123]. The little genetic divergence (4%) between SARS-CoV-2 and the nearest known bat coronavirus (bat-CoV) indicates the potential existence of an intermediate host. This organism may have enabled transmission from bats to humans by adapting the virus to a novel species [89,132].

According to many experts, bats are considered the natural reservoir of the SARS-CoV-2 virus [184]. However, for the virus to be transmitted from bats to humans, an intermediate host is required. In recently published studies—two genomic analyses—scientists suggested that snakes might be such an intermediate host. Another study identified a «special similarity» between the main protein of the novel CoV and the protein of HIV-1. However, following criticism from the scientific community, the authors removed the preprint from the scientific journal. Subsequent studies examined viromes obtained from pangolin lung, intestinal and blood tissues for similarities to SARS-CoV-2. The analysis found that the genome sequences of the pangolins studied corresponded to similar sequences of the SARS-CoV-2 virus in humans in the range of 85.5% to 92.4%. The pangolin is proposed as a potential intermediate host due to the identification of SARS-CoV-2-related CoVs in Malayan pangolins (*Manis javanica*) [130].

Moreover, disease detection, identification, prediction of interspecies transmission, research, early identification of infection foci, and the control and prevention of spread rely on international collaboration. The WHO underscores the significance of surveillance for COVID-19 and other respiratory viruses, as well as the necessity for data sharing among nations [185]. Taking this into account, during the pandemic, databases and nomenclature systems were created for the SARS-CoV-2 virus, such as the World Health Organization (WHO), which classified variants and assigned each lineage a name from the Greek alphabet [186]; the system for assigning phylogenetic lineages of global outbreaks (PANGOLIN, Pango); the Global Initiative on Sharing All Influenza Data (GISAID); as well as the Nextstrain platform, which enables real-time phylogenetic and phylodynamic analysis of SARS-CoV-2 variants to track their epidemiology and genetic evolution (Table 9) [136].

During the pandemic and continuing at present, the widespread circulation of the SARS-CoV-2 virus worldwide makes it critically important to maintain these systems and ensure timely data sharing in accordance with proper principles. In global monitoring of SARS-CoV-2 circulation, it is also essential to track its spread among animal populations and chronically infected individuals. These measures are key aspects of the global strategy aimed at reducing the emergence of mutations that may have negative consequences for public health [188].

Future coronavirus outbreaks are influenced by various factors and necessitate thorough assessments within the One Health paradigm. A comprehensive strategy encompassing epidemiological surveillance, ecosystem preservation, livestock management, research, and international collaboration is essential to mitigate the possibility of a new pandemic. The One Health concept promotes sustainable interaction among humans, animals, and ecosystems to prevent risks and manage their consequences.

## 10. Conclusions and Future Directions

In summary, the classification of CoVs, derived from molecular and phylogenetic analyses, identifies four genera: α-CoV, β-CoV, δ-CoV, and γ-CoV, which are part of the family *Coronaviridae*. These genera encompass numerous viruses that infect a range of mammalian and avian species, including humans. The genus β-CoV is linked to several significant diseases, including SARS-CoV, MERS-CoV, and SARS-CoV-2, which causes COVID-19.

Natural reservoirs play a significant role in the dissemination of CoVs, as they are organisms in which viruses can persist for extended durations without causing substantial harm to their hosts. The primary natural reservoirs of CoVs are bats, particularly for the genera α-CoV and β-CoV, and birds, which are associated with γ-CoV and δ-CoV. Bats are optimal hosts for the maintenance and dissemination of coronaviruses, attributable to their ecological and physiological diversity as well as their extensive geographical distribution.

Intermediate hosts are essential for moving CoVs from their natural reservoirs to humans. For instance, dromedary camels are known to be intermediate hosts for MERS-CoV, while palm civets have been linked to the spread of SARS-CoV. These animals act as biological «bridges» between reservoir species (like bats) and the final human host. This lets the virus adapt to other species before it spreads.

CoVs may infect a wide range of host species because they exploit a variety of cellular receptors, such as ACE2, DPP4, and APN, to do so. CoVs can cross species barriers because the way they interact with receptors is different. SARS-CoV, MERS-CoV, and SARS-CoV-2 are all examples of successful cross-species transmission that have led to zoonotic outbreaks. CoVs can also generate new strains of the virus that are very contagious in people because they mutate and combine so often.

While biological characteristics of the virus are fundamental to interspecies transmission, social and environmental factors also contribute significantly. Increased human–wildlife interaction, global trade in animals, intensive livestock production, and habitat destruction all facilitate the movement of viruses from animals to humans. These dynamics considerably heighten the risk of emerging infectious diseases, especially in the context of globalization, urbanization, and ecological disruption.

Molecular surveillance approaches should be implemented in an integrated framework to enable early detection of emerging threats and to prevent their dissemination. This involves the identification and sequencing of viral strains circulating in both animal and human hosts, as well as the analysis of viral evolutionary dynamics. The integration of data streams from public health and veterinary sectors allows for the timely recognition of high-risk sources before they spill over into human populations.

Looking ahead, a key priority is the incorporation of molecular and genomic findings into coordinated genomic monitoring systems. While many countries have established genomic surveillance programmes for viruses in human populations, systematic monitoring in animal hosts remains insufficient. Several research groups have highlighted the need to establish global repositories of CoV samples from animals and to maintain these databases in near real-time. Such infrastructures would enable earlier detection of evolutionary trajectories and potential intermediate strains at the human–animal interface.

Molecular studies and genomic surveillance of high-consequence pathogens such as CoVs in this context will play a critical role in future prevention strategies. A detailed understanding of viral structure, variability and biological properties provides the basis for rational vaccine design, antiviral drug development, and the creation of diagnostic assays with optimized sensitivity and specificity.

Another emerging concern involves environmental change and global urbanization, which are increasingly recognized as major drivers of the evolution and emergence of high-risk infectious diseases. Consequently, future research should prioritize the assessment of interactions among climate variables, biodiversity loss, wildlife migration patterns and human land-use practices using integrated modelling approaches. The One Health framework offers a comprehensive lens through which these interconnected factors can be analyzed as components of a shared risk landscape.

Thus, the One Health concept provides a strategic framework for integrating knowledge and resources to protect human, animal, and environmental health. A sustainable response to zoonotic threats can only be achieved through interdisciplinary collaboration and global coordination. Investing in scientific collaboration, data transparency, training, and early warning systems is the foundation for effectively managing future outbreaks of dangerous diseases.

Therefore, understanding the taxonomy, classification, and transmission pathways of CoVs from reservoirs to humans via intermediate hosts is crucial for predicting potential future outbreaks and developing effective policies aimed at mitigating potential pandemics. Further research into the evolution, ecology, and interspecies transmission mechanisms of viruses will enhance our preparedness and reduce the risk of new pandemics. Given the current biological instability, it is necessary to reconsider the objectives within national and international health systems. Understanding and predicting viral transmission mechanisms between species is fundamental to 21st-century biosafety, where One Health must be the cornerstone of scientific and political strategies.

## Figures and Tables

**Figure 1 microorganisms-14-00043-f001:**
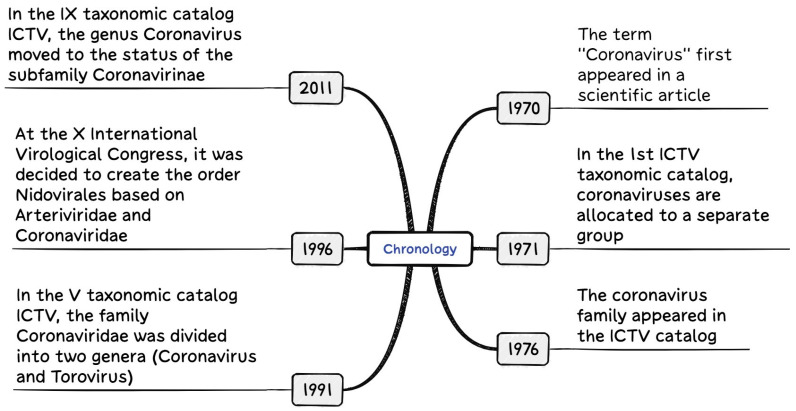
Chronology of the evolution of *Coronaviridae* family classification [3,5,6].

**Figure 2 microorganisms-14-00043-f002:**
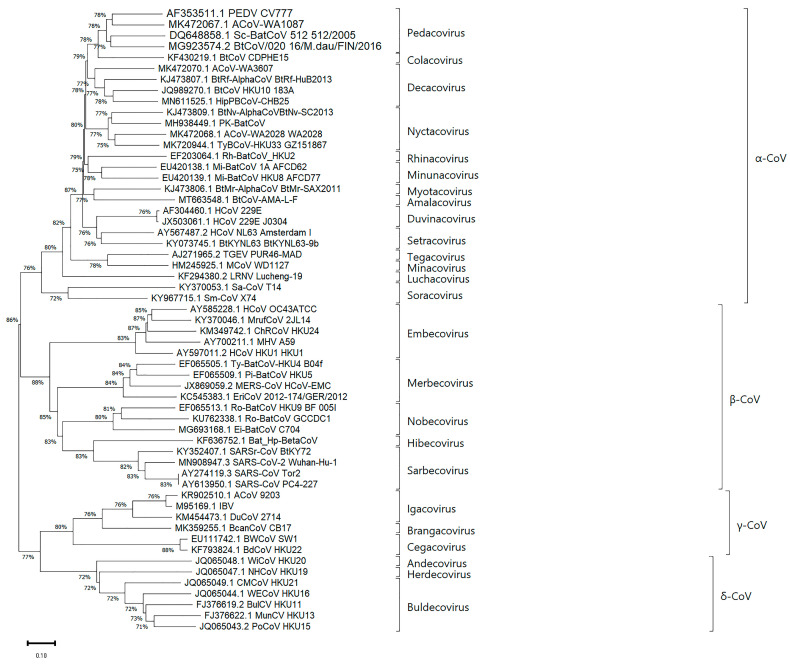
The current taxonomy of the *Coronaviridae* family [9].

**Figure 3 microorganisms-14-00043-f003:**
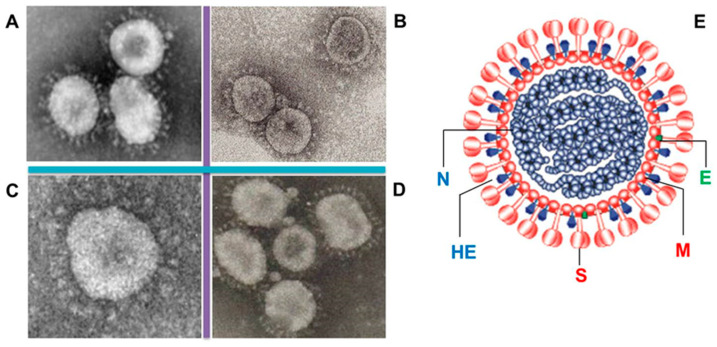
CoV morphology and structure. (**A**) Human coronavirus 229E (HCoV-229E), from C.M. Harrison et al. [15]; (**B**) Severe acute respiratory syndrome (SARS-CoV), from K. Stadler et al. [16]; (**C**) Porcine *deltacoronavirus*, from H. Hu et al. [17]; (**D**) Infectious bronchitis virus (IBV) [18]; (**E**) Schematic diagram of the coronavirus virion, from P. Woo et al. [19]. S—glycoprotein (red); E—envelope protein (green); M—membrane protein (red); N—nucleocapsid protein (blue); HE—haemagglutination and esterase (blue). A scale bar of 100 Nm is applicable only to (**B**,**C**). For (**A**,**D**), the scale bar is not presented in the referenced literature.

**Figure 4 microorganisms-14-00043-f004:**
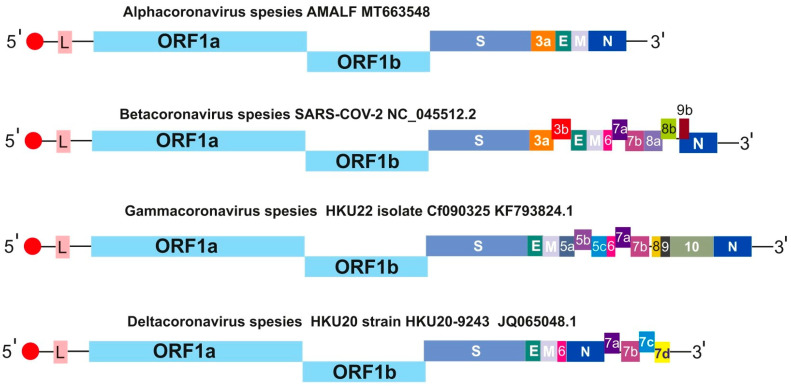
Genome structure and coding potential of human CoVs [19,23]. The genome structure with open reading frames (ORFs) is represented by the following colours: ORF1ab—sky blue; S—steel blue; ORF3a—orange; ORF3b—red; E—green; M—lavender, ORF5a—electric blue-purple, ORF5b—midnight purple; ORF5c—blue; ORF6—hot pink; ORF7a—indigo; ORF7b—purple, ORF7c—azure, ORF7d—yellow, ORF8—marigold; ORF8a—lilac; ORF8b—lime green; ORF9—dark; ORF9b—burgundy ORF10—moss green, N—cobalt.

**Figure 5 microorganisms-14-00043-f005:**
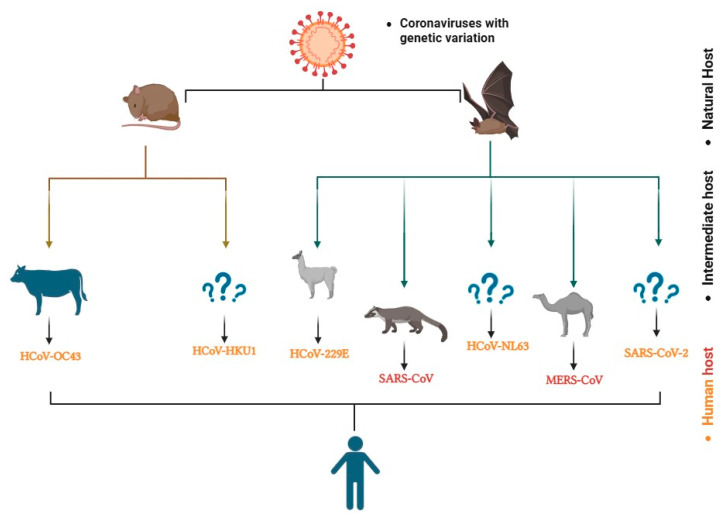
The animal origin of HCoV [107,108,109].

**Figure 6 microorganisms-14-00043-f006:**
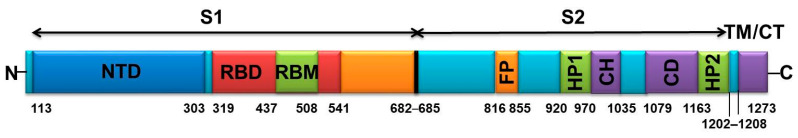
Schematic domain organization diagram of the spike protein gene [135]. The coloured bars describe the structural domain of the spiked protein. NTD, N-terminal domain (blue); CTD, C-terminal domain (red); RBD, receptor binding domain (red); RBM, receptor binding motif (green); HP1, heat protein 1 (green); HP2, heat protein 2 (green). Fusion peptides—(orange colour); TM—transmembrane domain (purple colour). CT—cytoplasm domain (purple colour).

**Figure 7 microorganisms-14-00043-f007:**
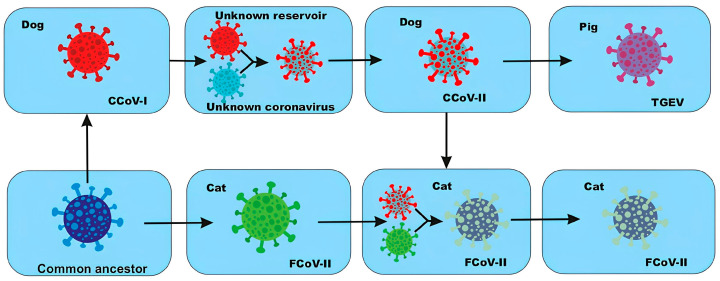
Interspecies transmission of CCoV and FCoV [170]. There is a hypothesis that feline coronavirus type I (FCoV-I) and canine coronavirus type I (CCoV-I) originated from a common ancestor. Subsequently, CCoV-I underwent recombination with an unknown coronavirus, resulting in the formation of a new variant—canine coronavirus type II (CCoV-II). Later, CCoV-II also recombined with FCoV-I, presumably in the body of an unknown host, leading to the emergence of feline coronavirus type II (FCoV-II). In addition, it is believed that CCoV-II may have been transmitted to pigs, contributing to the emergence of the transmissible gastroenteritis virus (TGEV).

**Figure 8 microorganisms-14-00043-f008:**
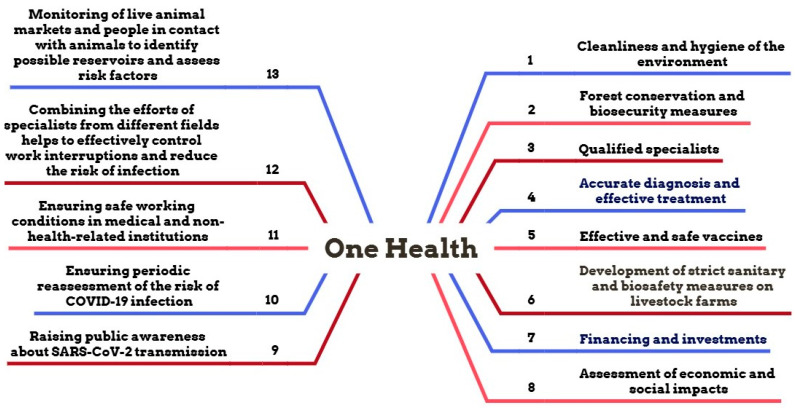
Unified public health measures to prevent the spread of infections and protect public health [92,174,175].

**Figure 9 microorganisms-14-00043-f009:**
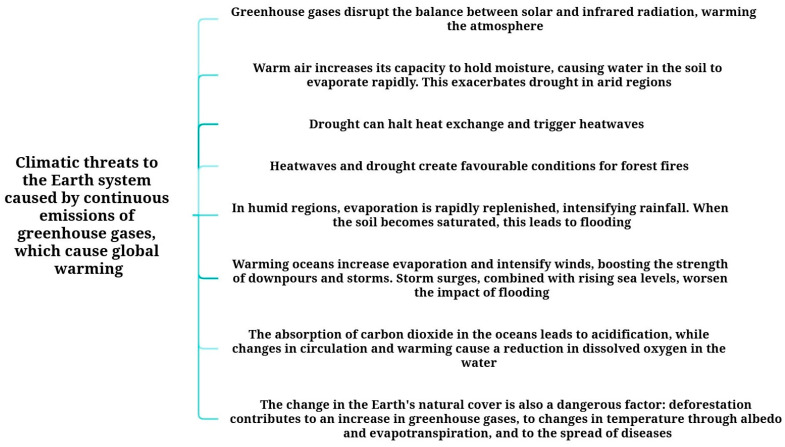
Ongoing greenhouse gas emissions associated with global warming introduce significant climate-related risks to the Earth system [181].

**Table 1 microorganisms-14-00043-t001:** CoVs detected in pigs: year of First Detection, geographical distribution, clinical signs and foci of infection.

Variety	Year of Emergence (Re-Emergence)/Distribution	Neonatal Piglet Mortality	Clinical Symptoms/Localization of Infection	Sources
TGEV (transmissible gastroenteritis coronavirus)	1946 (USA, America)/America, Europe, Asia	High	Diarrhea, catarrhal-hemorrhagic gastroenteritis/Small intestine	[24,25,26]
PRCV (porcine respiratory coronavirus)	1984 (Belgium)/Europe	Low	Dyspnea, tachypnea, anorexia, growth retardation/Respiratory tract	[27,28]
PHEV (porcine hemagglutinating encephalomyelitis virus)	1958 (Ontario, Canada)/Canada, Belgium, China, Argentina, South Korea, USA, Japan	Not specified	Vomiting, emaciation, limb tremors/central nervous system, Respiratory tract	[3,29,30]
Porcine epidemic diarrhea (PED)	1972 (England)/Belgium, Germany, England	High	Acute diarrhea (in fattening piglets, sows, piglets older than 10 weeks, and adult pigs)/Small intestine	[31]
Porcine epidemic diarrhea 2	1976 (England)/Netherlands, Germany, France, Bulgaria, Hungary and Switzerland		Acute diarrhea (in pigs of all ages)/Small intestine	[31,32]
PEDV (porcine epidemic diarrhea virus)	1978 (England and Belgium)/Europe			[31]
PDCoV (porcine *deltacoronavirus*)	2012 (Hong Kong, China)/China, Canada, South Korea, Thailand, Laos, Vietnam	Not specified	Diarrhea/Small intestine	[33]
SADS-CoV (swine acute diarrhea syndrome coronavirus)	2017 Guangdong, China	High	Anorexia, diarrhea, vomiting, dehydration, weight loss, lethargy/Small intestine	[24]

**Table 2 microorganisms-14-00043-t002:** CoVs affecting Felidae [34,35]. The table provides a summary of the primary CoVs identified in felines, including the time and location of initial detection as well as associated clinical manifestations.

Variety	Year of Emergence	Incubation Period	Clinical Symptoms	Susceptible Animals	Notes	Sources
Feline infectious peritonitis (FIP)	1963 USA	2 days–2 weeks	Exudative (moist, non-parenchymatous) type and Granulomatous lesion.	Domestic and wild felids: African lion, cougar, leopard, cheetah, jaguar, lynx, serval, caracal, sand cat, Pallas’s cat, Europe	Naturally occurring infection; humans are not carriers.	[34,35].
Feline enteric coronavirus (FECV)	Post-World War II	From ten days to several months	Mild gastrointestinal symptoms or asymptomatic carriage	Widespread, closely associated with FIP	Potential ancestor of FIPV; virulent form of FECV.	[34,35,37].
Feline coronavirus (FCoV)	1980s		Mild enteritis in cats, may exacerbate FIPV infection	Widespread among domestic cats	It is considered a virulent form of FECV; its epidemiology is closely linked to FIP	[34,35].

**Table 3 microorganisms-14-00043-t003:** CoVs affecting ferrets. The table summarizes the primary CoVs identified in ferrets, including the timeline, location of initial detection, and associated clinical signs.

Variety	Year of Emergence/Distribution	Clinical Symptoms	Tropism (Cellular/Organ-Specific)	Sources
Epizootic catarrhal enteritis virus (ECE)	1993 (USA)/USA and Europe	Lethargy, hyporexia, anorexia, vomiting	Intestinal (epithelial cells of the intestinal mucosa/gastrointestinal tract)	[3,46,47]
Ferret enteric coronavirus (FRECV)	No data available/USA	Diarrhea, lethargy, anorexia, vomiting	Intestinal (epithelial cells of the intestines/gastrointestinal tract)	[47]
Ferret systemic coronavirus (FRSCV)	2004 (Spain)/Netherlands, United Kingdom, Spain, USA, Peru, Japan	Diarrhea, weight loss, lethargy, hyporexia, anorexia, vomiting, sneezing, coughing, laboured breathing, nasal discharge, dehydration, bruxism, systolic heart murmur, jaundice, focal erythematous skin lesions, green urine, inflamed rectal mucosa, rectal prolapse	Intestinal, respiratory (epithelial cells of the respiratory tract/respiratory organs)	[47,48]

**Table 4 microorganisms-14-00043-t004:** Hosts and reservoirs of some animal CoVs with the indication of genus and subgenus.

Genus	Subgenus	Virus (Abbreviation)	Host	Reservoir	Sources
α-CoV	Tegacovirus	TGEV	Pig	Unknown	[92]
α-CoV	Tegacovirus	CCoV	Dog	Domestic dog (*Canis lupus familis*)	[93]
α-CoV	Tegacovirus	FIPV	Cat	Domestic cat (*Felis catus*)	[93]
α-CoV	Pedacovirus	PEDV	Pig	Bats (*Chiroptera*)	[94,95]
α-CoV	Rhinacovirus	SADS-CoV	Pig	Bats (*Chiroptera*)	[93]
α-CoV	Minacovirus	FRSCV	Ferret	Ferret (*Mustela putorius*)	[93]
α-CoV	Minacovirus	MCoV	Mink	Mink (*Mustela vison*/*Neovison vison*)	[96]
β-CoV	Embecovirus	PHEV	Pig	Birds	[97,98]
β-CoV	Embecovirus	CRCoV	Dog	Rodents Family Muridae	[93]
β-CoV	Embecovirus	BCoV	Cow	Dog/wild ruminants	[99]
β-CoV	Embecovirus	ECoV	Horse	Unknown	[100]
β-CoV	Sarbecovirus	SARS-like CoV	Raccoon dog	Bats (*Chiroptera*)	[101]
β-CoV	Merbecovirus	EriCoV	Hedgehog	Hedgehog	[102,103,104]

**Table 5 microorganisms-14-00043-t005:** Mutations in the spike protein of SARS-CoV-2 variants that may contribute to enhanced pathogenic properties.

No.	Mutation(s)	Spike Domain/Region	Impact on Viral Pathogenicity and Immune Evasion	Reference
1	K417N/T	RBD	Associated with immune escape; may reduce vaccine efficacy and contribute to reinfections	[136,137,138]
2	N439K	RBD	May facilitate immune evasion by reducing antibody neutralization efficacy	[137,139]
3	L452R	RBD	May disrupt interactions with residues I103 and V105, leading to reduced antibody neutralization	[136,137]
4	Y453F	RBD	Significantly enhances RBD binding affinity to hACE2, acting as a key determinant of viral pathogenesis	[137]
5	S477N	RBD	Enhances hACE2 binding and confers resistance to convalescent sera and neutralizing antibodies, increasing infectivity	[137]
6	T478K	RBD	Linked to immune escape and potential resistance to monoclonal antibody-based therapeutics	[136,137]
7	E484K/Q/A	RBD	Increases ACE2 binding affinity; promotes adaptive evolution and enhanced virulence	[136,137]
8	T478K, Q493K, Q498R	RBD	An increase in binding affinity between the RBD S protein and the ACE2 receptor is expected, which in turn may contribute to increased virus infectivity	[136,140]
9	N501Y	RBD	Contributes to immune escape and reduced antibody neutralization; plays a critical role in transmission and virulence	[137,138]
10	D614G	RBD	Increases spike density on virions and enhances ACE2 binding, leading to higher infectivity and transmissibility	[136,141,142]
11	Q677P/H	S1/S2 cleavage site proximity	May enhance cellular entry; however, increased transmissibility remains unconfirmed	[136]
12	P681R/H	S1/S2 cleavage site	Associated with improved viral entry into host cells	[143]
13	Δ69–70	NTD	Linked to immune escape	[136,138]
14	A222V	NTD	No significant functional impact reported	[137,144]

**Table 6 microorganisms-14-00043-t006:** Certain receptors of various CoVs.

Virus	Receptor	Reference	Virus	Receptor	Reference
Animal Coronaviruses			Human Coronavirus		
TGEV	APN	[1,91,105]	HCoV-229E	APN	[1,93]
PEDV	APN	[1,91]	HCoV-NL63	ACE2	[3,91,93,105]
PRCV	APN	[91,105]	HCoV-OC43	SIALIC ACIDS	[91,93,105]
CCoV	APN	[1,91,93,105]	HCoV-HKU1	SIALIC ACIDS	[91,93,105]
FIP	APN	[3,91,93]	SARS-CoV	ACE2	[3,90,91,93,105]
FCOV	APN	[105]	MERS-CoV	DPP4	[1,90,91,92,133]
FRSCV	Unknown	[93]	SARS-CoV-2	ACE2	[90,91,92,105,135]
SADS-CoV	Unknown	[93]			
BCoV	SIALIC ACIDS	[1,91]			
PHEV	SIALIC ACIDS	[105]			

**Table 7 microorganisms-14-00043-t007:** Comparative characterization of the main Bovine-like CoVs.

Animal/Virus Species	Status (Domestic/Wild)	Year of First Detection	Clinical Signs	Overall Nucleotide Identity with Other CoVs	Sources
Water buffalo (BCoV-like)	Domestic	1985, Bulgaria	Diarrhea	99% (BCoV)	[156]
Goats (CoV-like)	Domestic	2006, Turkey	Neonatal diarrhea	95% are identical to representatives of Embecovirus	[156,157,158,159]
Sheep (CoV-like)	Domestic	1978, Australia	Diarrhea	95% (BCoV)	[160,161]
Reindeer (*Cervidae*)					
Caribou/reindeer *(Cervidae)*	Wild	1978, Canada	Diarrhea	Unknown	[156]
Moose/elk	Wild	1991, Western North America	Diarrhea	99% (BCoV)	[156,162]
Sambar and white-tailed deer	Wild	1993–1994, USA	Severe diarrhea	99.6% (BCoV)	[156,161]
Sika deer	Wild	2006–2007, Japan	Diarrhea	Unknown	[156]
Water deer	Wild	2010–2012, South Korea	Respiratory disease	98% (BCoV)	[156,163,164]
Wild cattle					
Muskox (*Ovibox moschatus*)	Wild	1979–1980, UK	Diarrhea	Unknown	[156]
European bison (*Bison bonasus*)	Wild	2010, South Korea	Severe diarrhea	99.5% (BCoV)	[156,165]
Antelopes					
Waterbuck	Wild	1982, UK	Watery diarrhea	99.6% (BCoV)	[156,161]
Sitatunga	Wild	1979–1980, UK	Severe diarrhea	99.9% (BCoV)	[156,165]
Nyala	Wild	2010, South Korea	Diarrhea	99.9% (BCoV)	[156,165]
Black antelope	Wild	2003, USA	Diarrhea	99.6% (BCoV)	[156,161]
Other animals					
Giraffe (*Giraffa camelopardalis*)	Wild	2003, USA	Mild and severe diarrhea	99% (BCoV)	[156,165,166]
Himalayan tar (*Hemitragus jemlahicu*)	Wild	2010, South Korea	Weakness, depression, anorexia, bloody diarrhea, and dehydration	99% (BCoV)	[156,165]

**Table 8 microorganisms-14-00043-t008:** Comparative characterization of the main representatives of camelid CoVs.

Animal/Virus Species	Status (Domestic/Wild)	Year of First Detection	Clinical Signs	Overall Nucleotide Identity with Other CoVs	Sources
Old World camels					
Dromedary calf	Domestic	2002, USA	Diarrhea	Unknown	[156,168]
DcCoV UAE-HKU23 (dromedary)	Domestic	2013, UAE	Diarrhea	94.1% (BCoV)	[156,169]
New World camelids					
Alpaca CoV	Wild	1998, USA	Severe diarrhea	92.2% (HCoV-229E) and 99.5% (BCoV)	[114,156,160]

**Table 9 microorganisms-14-00043-t009:** Comparative scheme of classification systems for SARS-CoV-2 variants used by WHO, Pango, GISAID, and Nextstrain, indicating the date and place of first detection of each variant [85,136,187].

WHO Label	Pango Lineage	GISAID Clade/Lineage	Nextstrain Clade	Country of Origin
Alpha	B.1.1.7	GRY (formerly GR/501Y.V1)	20I	United Kingdom; Sep-2020
Beta	B.1.351	GH/501Y.V2	20H	South Africa; May-2020
Gamma	P.1	GR/501Y.V3	20J	Brazil; Nov-2020
Delta	B.1.617.2	G/478K.V1	21A/21I/21J	India; Oct-2020
Epsilon	B.1.427/B.1.429	GH/452R.V1	21C	United States of America; Mar-2020
Zeta	P.2	GR/484K.V2	20B	Brazil; Apr-2020
Eta	B.1.525	G/484K.V3	21D	Multiple countries; Dec-2020
Theta	P.3	GR/1092K.V1	21E	Philippines; Jan-2021
Iota	B.1.526	GH/253G.V1	21F	United States of America; Nov-2020
Kappa	B.1.617.1	G/452R.V3	21B	India; Oct-2020
Lambda	C.37	GR/452Q.V1	21G	Peru, Dec-2020
Mu	B.1.621	GH	21H	Columbia; Jan-2021
Omicron	B.1.1.529	GRA	21K, 21L, 21M, 22A–22F, 23A–23F, 24A–24I, 25A–25C	Southern African countries, Nov-2021

## Data Availability

No new data were created or analyzed in this study. Data sharing is not applicable to this article.

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
