# Peer review of "Alpha- and Beta-Coronaviruses in Humans and Animals: Taxonomy, Reservoirs, Hosts, and Interspecies Transmission"

_microorganisms, 2025, doi:10.3390/microorganisms14010043_

Round 1

Reviewer 1 Report

Comments and Suggestions for Authors

In this review entitled “Alpha- and beta-coronaviruses in humans and animals: taxonomy, zoonotic origin, natural reservoirs, intermediate hosts, receptors, and challenges within the One Health concept”, the author overviewed current knowledge on coronavirus taxonomy, major viral species, natural reservoirs, intermediate hosts, and mechanisms of interspecies transmission thoroughly. It also takes COVID-19 for example to highlight the importance of the One Health approach as an integrated, multidisciplinary strategy for monitoring, prevention, and control of coronavirus infections at the human – animal – environment interface . The manuscript could be published in Microorganisms.

Some minor revision:

  1. Table 5 and Table 6: the first line, suggestion: Animal/Virus species
  2. The heading order number of sections is incorrect.
  3. Line 862: RBD or S gene? RDB is part of S gene.
  4. Line 987: DNA or RNA?
  5. Line 1105: It seems that biosafety is more appropriate here.

Author Response

(Reviewer 1)

We sincerely thank the reviewer for the thoughtful comments and valuable suggestions, which have greatly helped to improve the quality and clarity of our manuscript.

Below, we provide a point-by-point response to each of the reviewer’s comments. All modifications have been carefully incorporated into the revised version of the manuscript.

  1. Table 5 and Table 6: the first line, suggestion: Animal/Virus species

In Tables 5 and 6, animal species were replaced with animal/virus species

  1. The heading order number of sections is incorrect.

Thank you for pointing this out. The erroneous section numbering has been corrected in the revised version of the manuscript

  1. Line 862: RBD or S gene? RDB is part of S gene.

The RBD constitutes a specific region within the S gene

  1. Line 987: DNA or RNA?

In the article, the word DNA was replaced with the word virome. This is because the authors of the reviewed paper had studied the virome in pangolin tissues.

  1. Line 1105: It seems that biosafety is more appropriate here.

The word biosecurity was replaced with the word biosafety

Reviewer 2 Report

Comments and Suggestions for Authors

This review article provides a comprehensive overview of alpha- and beta-coronaviruses, covering their taxonomy, structure, discovery in animals and humans, natural reservoirs, intermediate hosts, receptors, interspecies transmission, and implications within the One Health framework. The topic is timely and relevant, especially in the post-COVID-19 era, and the article effectively synthesizes a broad range of literature. The figures and tables are generally helpful for illustrating key concepts, such as genome organization, taxonomy, and host ranges.

However, the manuscript requires major revisions before publication. The primary issues are related to the English language (numerous grammatical errors, incorrect phrasing, and inconsistencies), structural organization (some sections feel disconnected or repetitive), and reference (many citations are pre-2023, with limited updates to 2025). 

Major Concerns:

  • Structure and Organization:
    • The manuscript jumps between topics; e.g., animal coronaviruses (1.2.1) are detailed but human ones (1.2.2) feel too superficial. 
    • Porcine coronaviruses get extensive coverage, while others (e.g., equine, beluga whale) are brief.
    • Future perspectives (Section 8) are underdeveloped: expand on emerging threats like new variants or climate change impacts.
  • Scientific Content:
    • Some statements are overgeneralized, e.g., "Bats are the primary natural reservoirs" (true for many, but specify for alpha/beta vs. gamma/delta). 
    • Gaps: Receptors (1.4) are well-described, but add more on mutations affecting binding (e.g., SARS-CoV-2 variants). One Health (1.6–9) could include more case studies or quantitative data on surveillance impacts.
    • Figures/Tables: Fig. 1–9 are good, but ensure all are original or properly cited (e.g., Fig. 6 from [116]). Add captions with more detail; e.g., explain colors in Fig. 7 explicitly.
    • Methods Absence: As a review, no methods section is needed, but briefly describe the literature search criteria (e.g., databases, date range) in the introduction for transparency.

Minor Issues:

      • Formatting: use consistent italics for genera (e.g., Betacoronavirus).
      • Abstract/Conclusion: Abstract is descriptive but lacks key findings; the conclusion repeats the introduction, making it more synthesizing.

Final Remarks

This manuscript has strong potential as a thorough review of coronaviruses. Address the language, structure, and update issues in a major revision to make it publication-ready. I recommend resubmission after revisions, with a response letter detailing changes.

Author Response

(Reviewer 2)

The manuscript jumps between topics; e.g., animal coronaviruses (1.2.1) are detailed but human ones (1.2.2) feel too superficial.

Porcine coronaviruses get extensive coverage, while others (e.g., equine, beluga whale) are brief.

We would like to clarify that all data presented in this section are based on peer-reviewed scientific literature and authoritative sources. The apparent difference in depth between subsections 5. (animal coronaviruses) and 6. (human coronaviruses) reflects the actual state of current knowledge and the scope of our manuscript’s focus.

Animal coronaviruses are indeed far more diverse—both in terms of viral species and host range—than human coronaviruses, which are limited to seven known types (including SARS-CoV-2). In contrast, numerous coronaviruses circulate in livestock, wildlife, and companion animals, often with significant economic, veterinary, and zoonotic implications. Among them, porcine coronaviruses (such as PEDV, TGEV, and PDCoV) have been studied extensively due to their high pathogenicity, rapid spread in swine populations, and substantial impact on global pork production. Swine are particularly susceptible to multiple coronavirus infections, which justifies the more detailed discussion in our review.

Future perspectives (Section 8) are underdeveloped: expand on emerging threats like new variants or climate change impacts.

We, Section 9 (‘One Health and Global Surveillance’) We have expanded Section 9 (‘One Health and Global Surveillance’) to include information on SARS-CoV-2 variants, the possible emergence of SARS-CoV-2, and the role of global genomic databases (e.g., GISAID, NCBI, PANGOLINE, NEXTRAIN) in detecting and monitoring viruses.

It was written in section 9.2 about climate change.

Scientific Content:

Some statements are overgeneralized, e.g., "Bats are the primary natural reservoirs" (true for many, but specify for alpha/beta vs. gamma/delta).

Our manuscript primarily focuses on alpha- and betacoronaviruses in domestic, wild animals, and humans within the One Health framework. While bats are key reservoirs, a thorough discussion of bat and rodent coronaviruses would require extensive additional analysis beyond the scope of this review. Accordingly, we have removed the preliminary data on bat coronaviruses to maintain focus and concisenes

Receptors (1.4) are well-described, but add more on mutations affecting binding (e.g., SARS-CoV-2 variants).

We have expanded Section 1.4 to include key S protein mutations (e.g., N501Y, E484K, L452R) from major SARS-CoV-2 variants and their effects on ACE2 binding and viral fitness

One Health (1.6–9) could include more case studies or quantitative data on surveillance impacts.

In accordance with your suggestion, section 9 of the manuscript has been amended to SARS-CoV-2 in the context of One Health: from potential emergence to genomic monitoring and its content has been expanded. This amendment clarifies the section's focus and highlights the comprehensive nature of the study.

Figures/Tables: Fig. 1–9 are good, but ensure all are original or properly cited (e.g., Fig. 6 from [116]). Add captions with more detail; e.g., explain colors in Fig. 7 explicitly.

In accordance with your suggestion, all images in the article have been sourced from open-source materials and cited with the relevant literature. The main images were reformulated and modified without using the originals, ensuring their meaning was not lost. In addition, comprehensive captions were added to all images; for example, in Figure 7 the significance of the colours was clearly described.

Methods Absence: As a review, no methods section is needed, but briefly describe the literature search criteria (e.g., databases, date range) in the introduction for transparency.

  1. Methods

The search across all databases (PubMed, Web of Science, Google Scholar, etc.) was conducted using the following keywords and terms: «coronavirus», «alpha-coronavirus», «beta-coronavirus», «SARS-CoV-2», «MERS-CoV», «bats», «One Health», etc. The selec-tion of terms was carried out according to thematic sections. Articles published between YEAR and YEAR in English were considered. Priority was given to peer-reviewed original studies and reviews, especially those addressing taxonomy, host range, receptor in-teractions, and zoonotic transmission. As a result, both original and review studies were selected. Particular attention was paid to publications in English whose titles, abstracts, and key topics met the selection criteria. The selected materials were analyzed in terms of their relevance to the objectives of this review.

Minor Issues:

Formatting: use consistent italics for genera (e.g., Betacoronavirus).

In the article, all taxonomic names, including the names of genera such as Betacoronavirus, are consistently italicised.

Abstract/Conclusion: Abstract is descriptive but lacks key findings; the conclusion repeats the introduction, making it more synthesizing.

Your comments have been taken into account, and the main scientific conclusions have been added to the abstract.

Final Remarks

This manuscript has strong potential as a thorough review of coronaviruses. Address the language, structure, and update issues in a major revision to make it publication-ready. I recommend resubmission after revisions, with a response letter detailing changes.

Comprehensive revisions have been made to the article concerning linguistic style, structure and updates. These changes were aimed at making the work's content clearer, more systematically scientific and ready for publication. The response letter to the review clearly detailed all the changes made.

Reviewer 3 Report

Comments and Suggestions for Authors

The manuscript is an interesting narrative review of alpha- and beta-coronaviruses in humans and animals. The review is focused on the taxonomy and classification of Coronaviridae, their genomic organization and structural proteins with special focus on the major alpha- and beta-CoVs of veterinary and human importance (FCoV, CCoV, BCoV, PEDV, HCoV-OC43, HCoV-229E, SARS-CoV, MERS-CoV, SARS-CoV-2, etc.). Authors refer to the natural reservoirs (especially bats and rodents) and intermediate hosts (civets, camels, pangolins, etc.), to the mechanisms of interspecies transmission, including receptor usage, to the public health responses to SARS-CoV-2, with a specific section on measures implemented in Kazakhstan and the role of One Health in surveillance, preparedness, and prevention of coronavirus outbreaksOverall it’s a very comprehensive, data-rich narrative review of coronavirus biology and ecology, with a clear One Health framing (especially in the later sections). The study is of high scientific relevance, the topic is timely and important and the combination of human, animal, and environmental perspectives is valuable. As a review, novelty is moderate but acceptable. The main “new” element is the integration of classical taxonomy, host range, receptor usage, and a Kazakhstan-specific case study under a One Health umbrella. The structure is in general good, but the text is dense and long (43 pages). Some sections are extremely detailed and could be streamlined. There are recurring English language issues, spacing around hyphens, and minor typographical problems throughout. The manuscript is presented as a review, but there is no clear description of how the literature was selected (no search or inclusion criteria), and the One Health concept is emphasized in the title but is not very systematically developed in the body (mainly a short conceptual paragraph + general comments on collaboration).

Major comments

My general recommendation is major review and the points to be addressed by the authors are the following:

Scope and focus vs title

  • The title promises: “taxonomy, zoonotic origin, natural reservoirs, intermediate hosts, receptors, and challenges within the One Health concept”
    but:
    • Taxonomy and host range are covered in great detail.
    • Receptors are treated, but somewhat scattered and more descriptive than integrative.
    • One Health challenges are not developed as a central analytical thread; instead, One Health appears mainly in:
      • The abstract (last sentences)
      • A short conceptual paragraph around future outbreaks and surveillance
      • Indirectly in the Kazakhstan section.

My suggestion regarding this point is to either strengthen the One Health component into a dedicated, substantial section (e.g., “One Health challenges and opportunities for coronavirus surveillance and control”) by clearing  subsections for human health, veterinary health, wildlife, and environment, concreting  examples of One Health-style interventions (joint surveillance, shared databases, integrated risk assessment) and offering, thus, a  more critical discussion of gaps (e.g., wildlife surveillance in LMICs, trade/market interfaces, occupational exposures). Alternatively a different approach would be to change the title of the paper accordingly as to  “Alpha- and beta-coronaviruses in humans and animals: taxonomy, reservoirs, hosts, and interspecies transmission” and keep One Health as an important framing but not the main advertised outcome.

Need for clearer review of methodology

For a review in Microorganisms, even a narrative one, readers usually expect at least a minimal description of how the literature was gathered.  Currently, there is no section indicating databases searched (PubMed, Scopus, Web of Science, GISAID, etc.), date range covered, inclusion/exclusion  criteria (e.g., only peer-reviewed articles; language restrictions; whether preprints were considered) and any prioritization of systematic reviews, primary data, etc. My suggestion for this point would be to add a short  subsection in the Introduction or immediately after it, for example: “Literature search strategy: We conducted a narrative review using PubMed and Web of Science with search terms ‘coronavirus’, ‘alpha-coronavirus’, ‘beta-coronavirus’, ‘SARS-CoV-2’, ‘MERS-CoV’, ‘bats’, ‘One Health’, etc. Articles published between YEAR and YEAR in English were considered. We prioritized peer-reviewed original research and reviews, particularly those addressing taxonomy, host range, receptor interactions, and zoonotic transmission.” Obviously, the authors don’t need a PRISMA diagram for a narrative review, but a paragraph clarifying the approach will strengthen scientific rigor and transparency.

Length, redundancy, and structure

The manuscript is 43 pages with numerous detailed descriptions (especially the subsections for individual animal CoVs).  Some animal-virus subsections are extremely detailed and somewhat repetitive (e.g Feline coronaviruses / FIP, multiple canine coronaviruses (CCoV, CRCoV), bovine, porcine etc). While this is useful information, it may be more than what is needed for one review that also has to cover human coronaviruses, receptors, and One Health framing. My suggestion would be to streamline animal virus subsections and merge highly similar subsections and focus on host range, clinical importance, known zoonotic potential (or lack thereof) and key knowledge gaps related to interspecies transmission. Also the authors could use tables and figures more strategically. For example a table could summarize virus species, host(s), clinical syndrome, known reservoirs, known or suspected intermediate hosts and main receptor(s). This would allow the authors to shorten the text and avoid repeating similar descritpions. The authors could consider  reorganizing major sections so that each major promise of the title has its own clearly visible section, e.g.:

  1. Introduction
  2. Taxonomy and classification of Coronaviridae
  3. Genomic organization and structural proteins
  4. Alpha- and beta-coronaviruses of animals
  5. Human alpha- and beta-coronaviruses
  6. Natural reservoirs and intermediate hosts
  7. Receptor usage and molecular determinants of host range
  8. One Health aspects and global surveillance
  9. Case study: Measures in Kazakhstan during SARS-CoV-2
  10. Conclusions and future directions

Right now some of these topics are partially intermingled.

Balance between global overview and Kazakhstan-specific section

The section “Actions implemented in Kazakhstan in response to the SARS-CoV-2 epidemic” is interesting and adds a unique regional perspective. However, it appears late and somewhat disconnected from the rest of the text and it seems more like a national report than an integrated part of a coronavirus review. My suggestion would be to frame this  this section as a One Health case study, explain why Kazakhstan is presented (e.g. unique surveillance infrastructure, wildlife interfaces, large livestock sector, etc.) and how the measures taken fit the One Health concept (collaboration of human health, veterinary, environmental sectors). Authors should consider shortening some administrative/normative details that do not strongly contribute to the scientific message and emphasize on the surveillance of SARS-CoV-2 in humans and animals, laboratory  capacities and genomic surveillance, coordination between ministries/institutions and the lessons learned that are generalizable to other countries.

Receptor usage and molecular determinants of host range

Authors discuss receptor interactions (e.g. ACE2, DPP4, APN/CD13, etc.) in several places, but this information is somewhat scattered. I would suggest to make the following changes, given that “receptors” are in the title:

  1. Create a dedicated subsection on receptor usage for each major human and key animal CoV, clearly state:
    • Primary receptor
    • Key co-receptors / attachment factors if relevant
    • Any known species differences in receptor sequence/structure that influence host range (e.g. ACE2 residues that differ between bats, pangolins, humans).
  2. Integrate discussion of how mutations in the S protein and receptor-binding domain (RBD) have shaped past cross species transmissions (SARS-CoV, MERS-CoV, SARS-CoV-2), and may influence future ones.
  3. A summary figure or table mapping viruses to receptors and hosts would be very valuable for readers.

One health section

The One Health perspective is mainly expressed through general statements about the need for surveillance and early detection, for international cooperation and data sharing (WHO, GISAID, GenBank), whereas environmental and wildlife aspects mentioned briefly. In order to match the title authors should:

  1. Discuss specific interfaces where human–animal–environment interactions drive coronavirus emergence, e.g. wildlife trade and live animal markets, intensive livestock systems, companion animals in close contact with humans and occupational exposure (farm workers, veterinarians, wildlife handlers).
  2. Highlight concrete One Health initiatives, such as joint human-veterinary surveillance networks, integrated risk assessments across ministries and cross-sectoral outbreak investigations (e.g. MERS in camels/humans).
  3. Address challenges such as data sharing barriers between sectors/countries, underfunded wildlife surveillance and lack of standardized protocols in animal surveillance compared with human surveillance.

Minor comments

English language and style

The English is understandable, but many sentences could be made more natural and concise. Recurrent issues:

  • Spacing around hyphens and dashes:
    • g. “Alpha - and beta -coronaviruses”, “β -CoV”, “COVID -19”.
      → Should be “alpha- and beta-coronaviruses”, “β-CoV”, “COVID-19”.
  • Inconsistent capitalization:
    • “One health” vs “One Health”, “Coronavirus” vs “coronavirus”, etc.
      → Choose one consistent style.
  • Minor grammatical issues:
    • “Considered a virulent form of FECV; epidemiology closely linked to FIP.” could be “It is considered a virulent form of FECV; its epidemiology is closely linked to FIP.”
    • “Such investigations should employ both classical methodologies and contemporary molecular techniques, so facilitating…” → “…techniques, thereby facilitating…”
  • Some very long sentences could be split for ease of reading.

Suggestion:
Have the manuscript carefully edited by a fluent English speaker or a professional editing service, with special attention to:

  • Hyphenation and spacing
  • Subject-verb agreement
  • Run-on sentences
  • Consistent tense (mostly past vs present for general facts).

References and citation style

  • The reference list is extensive and generally appropriate, but:
    • Make sure all references are formatted exactly according to Microorganisms style (journal names, volume, issue, pages, DOI).
    • Check for truncation errors – some lines in the PDF preview appear split, e.g. author names broken, ellipses, etc. This may be a PDF artifact, but please carefully check the final bibliography in the manuscript file.
  • Consider ensuring that recent literature up to at least 2023/2024 is included, especially for:
    • SARS-CoV-2 variants and evolution
    • Novel bat and rodent CoV discoveries
    • One Health and coronavirus surveillance frameworks.

Figures and tables

  • Figure 3 (genome structure):
    • The color legend is detailed, but in print, too many similar shades may be hard to distinguish.
    • Consider simplifying the color scheme or grouping ORFs into functional categories (replicase, structural, accessory).
  • Figure 4 (chronology of classification):
    • Very useful historically.
    • Make sure the text is legible at journal column width; avoid overly small fonts.
  • Tables of virus properties (e.g. table with FCoV, CCoV, etc.):
    • Already helpful; you might expand these rather than long textual descriptions.
    • Check alignment and consistency of spelling (e.g. “coronavir us” vs “coronavirus” – there are some line-break artifacts).

Terminology and clarity

  • In the Introduction, the list of pathogens causing respiratory disease is very long (viruses, bacteria, parasites, fungi). While correct, it slightly distracts from the core focus on coronaviruses.
    • Consider shortening or summarizing this list to keep the narrative focused.
  • When introducing abbreviations (e.g. FIPV, TGEV, PEDV, etc.), ensure they are defined at first mention and used consistently thereafter.
  • When describing transmission routes, clearly distinguish:
    • Respiratory vs fecal–oral transmission
    • Direct vs indirect transmission
    • Proven vs suspected routes (especially in animals where data are limited).

General comments

Introduction: The Introduction sets the stage for emerging infections and respiratory diseases; however, the authors could shorten the general part on all pathogens, and transition faster to coronaviruses and mention  the motivation for this review (What gap does it fill compared with existing coronavirus reviews?, E.g. “Unlike previous reviews that focus primarily on SARS-CoV-2, we integrate human and veterinary perspectives, detail natural reservoirs and intermediate hosts, and discuss One Health challenges with a case study from Kazakhstan.”)

Sections on individual animal coronaviruses: For each major animal CoV, consider emphasizing whether there is evidence or potential for zoonotic transmission to humans., how it fits into the broader coronavirus evolution story (e.g. relationships to human CoVs, recombination events). Some detailed epidemiological numbers and historical descriptions could be shortened unless they directly support the One Health or interspecies transmission narrative.

Sections on human coronaviruses (HCoVs, SARS, MERS, SARS-CoV-2): These sections are generally well-structured and informative. Please make sure that SARS-CoV-2 discussion reflects current understanding of variants, reservoir candidates, and reverse zoonosis (e.g. infections in mink, white-tailed deer, etc.), as this directly relates to One Health, and the  MERS-CoV section clearly indicates the ongoing zoonotic risk from dromedary camels and why it remains a concern despite limited human-to-human spread compared with SARS-CoV-2.

Section on Kazakhstan response: As noted above, frame this clearly as a One Health case study. Authors should provide a short schematic or table summarizing main measures in human health, measures in animal health, environmental/public health measures and Responsible institutions and how they coordinated.

Conclusions: The conclusions already highlight the importance of surveillance and cooperation. The authors could strengthen them by kisting 3–5 key messages, for example:

  • Alpha- and beta-CoVs have a broad host range with bats and rodents as key reservoirs.
  • Molecular determinants such as receptor usage and S protein variability drive host jumps.
  • Integrated One Health surveillance is essential to detect and mitigate future coronavirus emergence.
  • National examples (such as Kazakhstan) show how cross-sectoral collaboration can be implemented in practice.
  • Major gaps remain in wildlife surveillance, data sharing, and preparedness in low-resource settings.

Author Response

(Reviewer 3)

Scope and focus vs title

Alpha- and beta-coronaviruses in humans and animals: taxonomy, reservoirs, hosts, and interspecies transmission

Need for clearer review of methodology

The search across all databases (PubMed, Web of Science, Google Scholar, etc.) was conducted using the following keywords and terms: «coronavirus», «alpha-coronavirus», «beta-coronavirus», ‘SARS-CoV-2’, «MERS-CoV», «bats», «One Health», etc. The selection of terms was carried out according to thematic sections. Articles published between YEAR and YEAR in English were considered. Priority was given to peer-reviewed original studies and reviews, especially those addressing taxonomy, host range, receptor interactions, and zoonotic transmission. As a result, both original and review studies were selected. Particular attention was paid to publications in English whose titles, abstracts, and key topics met the selection criteria. The selected materials were analyzed in terms of their relevance to the objectives of this review.

Length, redundancy, and structure

We appreciate the reviewer’s suggestion regarding the structure and clarity of the manuscript. In response, we have shortened several sections of the text to improve conciseness and relocated content to more appropriate parts of the manuscript in accordance with the logical flow of the article. The overall structure now better reflects the outline originally proposed in the manuscript plan.

Balance between global overview and Kazakhstan-specific section

This section has been removed from the manuscript, as it was deemed somewhat tangential to the main topic. Additionally, given that the original version of the paper was already quite lengthy (43 pages), we decided to exclude this part to maintain focus and conciseness in line with journal guidelines. However, we are happy to reinstate this section if the reviewer considers it essential for the completeness of the manuscript

Receptor usage and molecular determinants of host range

In this section, we have removed redundant or non-essential phrases that did not affect the overall meaning of the sentences or the text. Additionally, we have specified the respective viruses associated with each receptor to enhance clarity and scientific accuracy.

One health section

This section outlines key One Health–related risks, particularly the potential for cross-species disease transmission. It reviews current knowledge on the emergence and community spread of SARS-CoV-2, including its probable zoonotic origin. Existing One Health–oriented studies on possible transmission routes are summarized, and international guidelines and data sources on viral surveillance and monitoring are referenced

Minor comments

Spacing around hyphens and dashes: Your suggestion has been corrected/implemented wherever possible

Inconsistent capitalization: Your suggestion has been corrected/implemented wherever possible

Minor grammatical issues: In response to the reviewer’s comment, the text has been corrected and moved to a more suitable location within the manuscript

Figures and tables

Figure 3 (genome structure): The figure has been carefully revised and updated in accordance with your comment

Figure 4 (chronology of classification): The figure has been re-examined and redesigned based on your valuable feedback

Round 2

Reviewer 2 Report

Comments and Suggestions for Authors

The manuscript is now suitable for publication.

Author Response

We sincerely appreciate the reviewer’s comment regarding the clarity of the English. Several colleagues with strong English proficiency have carefully checked the manuscript and consider the language appropriate for publication, so we believe no major changes are required. At the same time, to ensure full compliance with the journal’s standards, we remain open to minor editorial adjustments and are willing to use the journal’s English editing service if recommended. We kindly ask the editor to advise us on the preferred course of action.

Reviewer 3 Report

Comments and Suggestions for Authors

All my comments have been addressed and the manuscript has been substantially improved. 

Author Response

(The authors gave the same response as above.)
